# Companies inadvertently fund online misinformation despite consumer backlash

Wajeeha Ahmad[1 ✉], Ananya Sen[2], Charles Eesley[1] & Erik Brynjolfsson[3]

The financial motivation to earn advertising revenue has been widely conjectured to be pivotal for the production of online misinformation[1-4]. Research aimed at mitigating misinformation has so far focused on interventions at the user level[5-8], with little emphasis on how the supply of misinformation can itself be countered. Here we show how online misinformation is largely financed by advertising, examine how financing misinformation affects the companies involved, and outline interventions for reducing the financing of misinformation. First, we find that advertising on websites that publish misinformation is pervasive for companies across several industries and is amplified by digital advertising platforms that algorithmically distribute advertising across the web. Using an information-provision experiment[9], we find that companies that advertise on websites that publish misinformation can face substantial backlash from their consumers. To examine why misinformation continues to be monetized despite the potential backlash for the advertisers involved, we survey decision-makers at companies. We find that most decision-makers are unaware that their companies' advertising appears on misinformation websites but have a strong preference to avoid doing so. Moreover, those who are unaware and uncertain about their company's role in financing misinformation increase their demand for a platform-based solution to reduce monetizing misinformation when informed about how platforms amplify advertising placement on misinformation websites. We identify low-cost, scalable information-based interventions to reduce the financial incentive to misinform and counter the supply of misinformation online.

The prevalence of online misinformation can have important social consequences, such as contributing to greater fatalities during the COVID-19 pandemic[10], exacerbating the climate crisis[11], and sowing political discord[12]. Yet the supply of misinformation is often financially motivated. The economic incentive to produce misinformation has been widely conjectured by academics and practitioners to be one of the main reasons websites that publish misinformation (hereafter referred to as 'misinformation websites' or 'misinformation outlets'), masquerading as legitimate news outlets, continue to be prevalent online[1-4]. During the 2016 US Presidential election, one operator of a misinformation outlet openly stated "For me, this is all about income"[13].

Media reports have anecdotally observed that companies and digital platforms contribute towards financially sustaining misinformation outlets via advertising[14,15]. Advertising companies can either place their advertisements directly on specific websites or use digital advertising platforms to distribute their advertisements across the internet (Methods, 'Background on digital advertising'). The vast majority of online display advertising today is done via digital advertising platforms that automatically distribute advertisements across millions of websites[16], which may include misinformation outlets. According to a recent industry estimate, for every US$2.16 in digital advertising revenue sent to legitimate newspapers, US advertisers send US$1 to misinformation sites[17].

Existing work to counter the proliferation of misinformation online has primarily focused on empowering news consumers[3,5] in order to reduce the demand for misinformation through interventions such as fact-checking news articles[6], providing crowd-sourced labels[8] and nudging users to share more accurate content[7]. However, a vital question remains regarding how the incentive to produce or supply misinformation may be countered. Indeed, recently, academics have proposed 'supply-side' policies for steering platforms away from the revenue models that might contribute towards sustaining harmful content[18]. Digital platforms have also attempted to decrease advertising revenue going to some misinformation websites[19]. However, despite these attempts, advertising from well-known companies and organizations continues to appear on misinformation websites, thereby financing such outlets[20,21]. Moreover, the supply of misinformation is expected to increase with generative AI technologies making it easier to create large volumes of content to earn advertising revenue[22,23].

In this Article, we attempt to provide a first step in understanding how to limit the financing of online misinformation via advertising using descriptive and experimental evidence. To tackle the problem of financing online misinformation, it is important to first understand

[1]Department of Management Science and Engineering, Stanford University, Stanford, CA, USA. [2]Heinz College of Information Systems and Public Policy, Carnegie Mellon University, Pittsburgh, PA, USA. [3]Institute for Human-Centered Artificial Intelligence, Stanford University, Stanford, CA, USA. ✉e-mail: wajeehaa@stanford.edu

the role of different entities within this ecosystem. In particular, we need to establish whether companies directly place advertisements on misinformation outlets or do so by automating such placement through digital advertising platforms. Although several mainstream digital platforms generate the vast majority of their revenue via advertising[3], little is understood about the role of advertising-driven platforms in financing misinformation. To evaluate the relative roles of advertising companies and digital advertising platforms in monetizing misinformation, we construct unique large-scale datasets by combining data on websites publishing misinformation with advertising activity per website over a period of three years.

Next, the extent to which companies can be dissuaded from advertising on misinformation websites depends on how their customers respond to information about the prevalence of companies' advertising on such websites. As people find out about companies advertising on misinformation websites through news and social media reports[20,24], they may reduce their demand for such companies or voice concerns against such practices online[25,26]. Therefore, it is important to measure the preferences of the people who consume a company's products or services regardless of whether these consumers visit misinformation websites themselves. To measure these effects, we conducted a survey experiment with a sample of the US population by randomly varying the pieces of factual information we provided to participants. By simultaneously measuring how people shift their consumption and the types of actors (that is, advertisers or digital advertising platforms) that they voice concerns about, we capture how peoples' reactions change as the degree to which advertisers and advertising platforms are held responsible varies. We also study how consumer responses may vary depending on the intensity of a company's advertising on misinformation websites by providing company rankings on this dimension.

Finally, whether decision-makers within companies are aware of their company's advertisements appearing on misinformation outlets and prefer to avoid doing so can have an important role in curbing the financing of misinformation. In recent years, advertisers have often participated in boycotts of advertising-driven platforms such as YouTube, Facebook and Twitter for placing their advertisements next to problematic content[27,28]. However, there is little systematic measurement of the knowledge and preferences of key decision-makers within companies in this context. To address this gap, we surveyed executives and managers by contacting the alumni of executive education programmes. Moreover, we conducted an information-provision experiment to examine whether decision-makers would increase their demand for a platform-based solution to avoid advertising on misinformation outlets when informed about the role of digital advertising platforms in monetizing misinformation.

We report three sets of findings from our descriptive and experimental analyses. First, our descriptive analysis suggests that misinformation websites are primarily monetized via advertising revenue, with a substantial proportion of companies across several industries appearing on such websites. We further show that the use of digital advertising platforms amplifies the financing of misinformation. Second, we find that people switch consumption away from companies whose advertising appears on misinformation outlets, reducing the demand for such companies. This switching effect persists even when consumers are informed about the role of digital advertising platforms in placing companies' advertisements on misinformation websites and the role of other advertising companies in financing misinformation. Third, our survey of decision-makers suggests that most of them are ill-informed about the roles of their own company and the digital advertising platforms that they use in financing misinformation outlets. However, decision-makers report a high demand for information on whether their advertisements appeared on misinformation outlets and solutions to avoid doing so. Those who were uncertain and unaware about where their advertising appeared also increased their demand for a platform-based solution to reduce advertising

on misinformation websites upon learning how platforms amplify advertising on such websites.

In sum, our results indicate that there is room to decrease the financing of misinformation using two low-cost, scalable interventions. First, improving transparency for advertisers about where their advertisements appear could by itself reduce advertising on misinformation websites, especially among companies who were previously unaware of their advertisements appearing on such outlets and were thus inadvertently financing misinformation. Second, although it is currently possible for consumers to find out about advertising companies financing misinformation through news and social media, platforms could make advertising on misinformation outlets more easily and continuously traceable to the advertising companies involved for consumers. Our results suggest that both simple information disclosures and comparative company rankings can reduce consumer demand away from companies advertising on misinformation websites.

We build on prior work analysing the ecosystem supporting misinformation websites[29–33] and programmatic advertising[34] by matching millions of instances of advertising companies appearing across thousands of news outlets with data on misinformation websites, thereby providing large-scale evidence of the ecosystem that sustains online misinformation over a consistent period of three years. Additionally, we present descriptive evidence about the relative roles of advertising companies and digital advertising platforms in financing misinformation. Next, our information-provision experiments examine the effects of advertising on misinformation websites for companies and platforms. Previous work has examined the conditions under which people react against companies for failing to operate up to their expectations—for example, due to service quality deterioration[26], not fulfilling social responsibilities[35], advertising next to violent content[36], or taking a political stance[37,38]. Our research design contributes to this literature in two key ways by: (1) measuring both types of potential consumer responses—that is, 'exit' and 'voice'—that are theorized in the literature[25]; and (2) doing so using incentive-compatible behavioural outcomes at the individual level, which enables us to capture costly decisions people make and move beyond stated preferences recorded in related experimental research[36,39]. More broadly, our research suggests an alternative approach to countering misinformation online by suggesting how the monetization of misinformation could be curbed using information interventions. Our study complements and extends prior work on using disclosures[40,41] and interventions to counter misinformation[5,7] by showing that disclosures about companies advertising on misinformation outlets can shift consumption away from such companies, ultimately incentivizing companies to reduce the financing of misinformation via advertising.

## Collection of website and advertising data

To categorize whether a website contains misinformation, we compiled a list of misinformation domains using three different sources: NewsGuard, the Global Disinformation Index (GDI) and websites used in prior work (see Methods, 'Collecting website data'). NewsGuard and the GDI use automated and manual methods to source and evaluate websites, but each website is rated manually by expert professionals who apply journalistic standards to evaluate online news outlets in a non-partisan and transparent manner.

We collected data on advertiser behaviour from 2019 to 2021 via Oracle's Moat Pro platform, which includes data collected by 'crawling' approximately 10,000 websites daily to create a snapshot of the advertising landscape. Moat's web crawlers mirror a normal user experience and attempt to visit a representative sample of pages for each website at least once a day. To the best of our knowledge, these data are the gold standard used by many industry stakeholders for competitive analysis. For all the websites in our sample that get non-zero traffic throughout this period and have advertising data available on

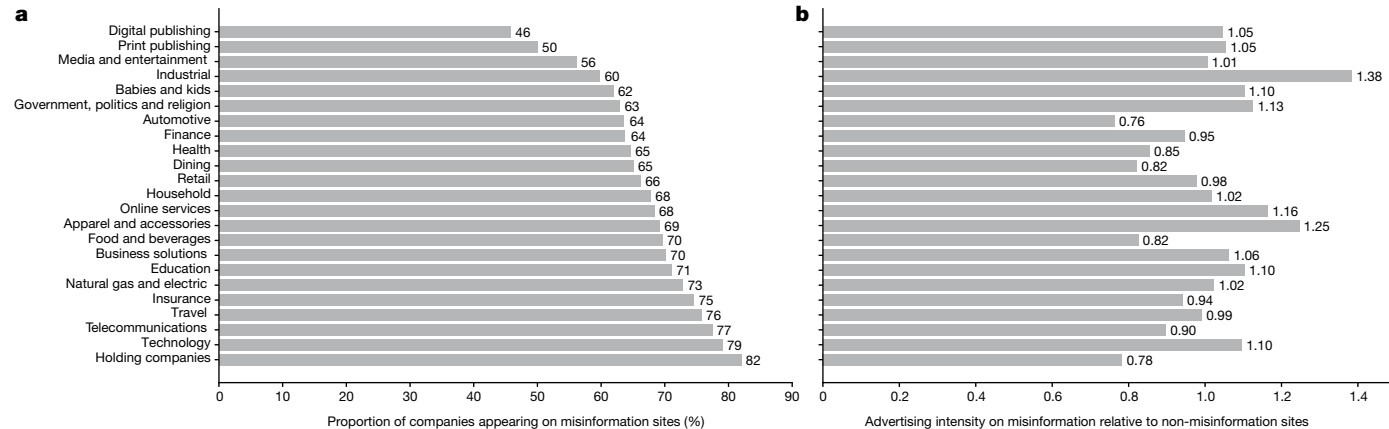

**Fig. 1 | Advertising companies appearing on misinformation websites by industry.** From 2019 to 2021, we recorded the number of times companies in a given industry appeared on the 5,485 websites in our sample per month. Our final sample of advertisers consists of 42,595 companies and 9,539,847 instances of companies advertising on the websites in our sample. We removed industries where the number of advertising appearances by all companies combined was below the 5th percentile of the total number of advertising appearances, resulting in a total of 23 industries. **a**, The proportion of companies in each industry that appear on misinformation websites at least once in our sample. **b**, The advertising intensity on misinformation sites relative to non-misinformation websites for each industry. This is calculated by dividing the proportion of advertisements from companies of that industry appearing on misinformation websites among all advertising appearances on misinformation websites with the same proportion for non-misinformation websites per industry. Therefore, values lower than 1 indicate less, values close to 1 represent similar and values higher than 1 represent greater advertising intensity on misinformation sites relative to non-misinformation websites.

the Moat Pro platform, we collected monthly data on the advertising companies appearing on each website and digital advertising platforms used by each website.

Our final dataset, which contains data on advertising and misinformation, consists of 5,485 websites (including 1,276 misinformation websites and 4,209 non-misinformation websites) and 42,595 unique advertisers with 9,539,847 instances of advertising companies appearing on news websites between 2019 and 2021. Additionally, for the most active 100 advertisers each year, as identified by Moat Pro, we collected weekly data on the websites that they appeared on and the digital advertising platforms that they used.

## Descriptive analysis

Of the websites in our sample, 89.3% were supported by advertising revenue between 2019 and 2021, and the majority of misinformation websites (74.5%) were monetized by advertising during this period. Moreover, among websites rated by NewsGuard, a much smaller percentage of misinformation websites had a paywall (2.7% in the USA and 3.2% globally) relative to non-misinformation websites (25.0% in the USA and 24.0% globally), which indicates a greater reliance on advertising for financing relative to other subscription-based business models among misinformation websites. Although different entities may have specific ideological or financial motivations for propagating online misinformation, data from NewsGuard-rated websites (see Supplementary Table 3) shows that relative to non-misinformation websites, misinformation websites were also more likely to be operated by individuals as opposed to corporate, non-profit or government entities. Given that advertising appears to be the dominant business model that sustains misinformation outlets, it merits a closer look. We find that companies that advertise on misinformation websites span a wide range of industries (Supplementary Table 4) and account for 46% to 82% of overall companies in each industry (Fig. 1a). These include several well-known brands among commonly used household products, technology products and business services, as well as finance, health, government and educational institutions among other industries. Further, the intensity of advertising on misinformation sites is similar (mean = 1.01, 95% confidence interval [0.945, 1.074], $t(22) = 0.311$, $P = 0.759$ from one-sample $t$-test, $n = 23$)

to that on non-misinformation sites for companies across several industries (Fig. 1b).

Next, we examined the role of digital advertising platforms in financing misinformation. For the one hundred most active advertisers in each year, we collected weekly data on the websites their advertisements appeared on and their use of digital advertising platforms. On average, about 79.8% of advertisers that used digital advertising platforms in a given week appeared on misinformation websites that week. In contrast, among companies that did not use digital advertising platforms in a given week, only 7.74% appeared on misinformation websites on average in a given week (two-sided $t$-test $t(192.12) = 93.903$, $P < 0.001$, $n = 144$). In other words, companies that used digital advertising platforms were approximately ten times more likely to appear on misinformation websites than companies that did not use digital advertising platforms. Moreover, we account for industry and time trends to find that the use of digital advertising platforms by companies substantially amplifies the likelihood of a company's advertising appearing on misinformation websites (see Extended Data Table 1).

## Effects of advertising on misinformation

Next, our survey experiment aimed to determine potential changes in consumer behaviour based on experimentally varied information about the roles of companies and platforms in financing misinformation via advertising. Using the framework of Hirschman[25], we measured how people (1) exit (that is, decrease their consumption), and (2) voice concerns about company or platform practices via online petitions in response to the information provided in an incentive-compatible manner.

## Average treatment effects

As detailed in Methods, 'Consumer experiment design', participants in our experiment were offered a gift card from a company of their choice. Our primary pre-registered outcome is whether respondents exit by switching their top gift card choice after receiving an information treatment, which takes the value one for people who switch and the value zero for all other participants ($n = 4,039$). To observe exit outcomes, we focus on company-related information treatments

## Table 1 | Average treatment effects on exit

| | Switch in preference | | Switch to lower preference | | Switch in category | | Switch to lower misinformation | |
|---|---|---|---|---|---|---|---|---|
| | 1 | 2 | 3 | 4 | 5 | 6 | 7 | 8 |
| Company (T1) | 0.13*** | 0.13*** | 0.08*** | 0.08*** | 0.05*** | 0.05*** | 1.03** | 0.69* |
| | (0.01) | (0.01) | (0.01) | (0.01) | (0.01) | (0.01) | (0.48) | (0.38) |
| | <0.001 | <0.001 | <0.001 | <0.001 | <0.001 | <0.001 | 0.031 | 0.075 |
| Platform (T2) | 0.03*** | 0.03** | 0.01 | 0.01 | 0.02* | 0.01 | 0.52 | 0.23 |
| | (0.01) | (0.01) | (0.01) | (0.01) | (0.01) | (0.01) | (0.54) | (0.48) |
| | 0.010 | 0.012 | 0.114 | 0.118 | 0.076 | 0.130 | 0.335 | 0.629 |
| Company and platform (T3) | 0.10*** | 0.10*** | 0.06*** | 0.06*** | 0.04*** | 0.04*** | 0.69 | 0.28 |
| | (0.01) | (0.01) | (0.01) | (0.01) | (0.01) | (0.01) | (0.49) | (0.38) |
| | <0.001 | <0.001 | <0.001 | <0.001 | <0.001 | <0.001 | 0.157 | 0.452 |
| Company ranking (T4) | 0.08*** | 0.08*** | 0.06*** | 0.06*** | 0.03*** | 0.02** | 1.57*** | 0.95** |
| | (0.01) | (0.01) | (0.01) | (0.01) | (0.01) | (0.01) | (0.50) | (0.39) |
| | <0.001 | <0.001 | <0.001 | <0.001 | 0.006 | 0.015 | 0.002 | 0.015 |
| Controls | No | Yes | No | Yes | No | Yes | No | Yes |
| Control group mean | 0.04 | 0.04 | 0.02 | 0.02 | 0.03 | 0.03 | 0.65 | 0.65 |
| Observations | 4039 | 4039 | 4039 | 4039 | 4039 | 4039 | 430 | 430 |

Ordinary least squares (OLS) regression results for each of the four treatment groups (T1, T2, T3 and T4) in our sample. Columns are numbered along the top. In columns 1 and 2, the dependent variable is a binary variable that takes the value 1 when a participant switches their gift card choice from their top choice company after receiving the information treatment and is zero otherwise ($n=4,039$). In columns 3 and 4, the dependent variable is a binary variable that takes the value 1 when a participant switches their gift card choice from their top choice company to a company they prefer less (as measured by how participants assign weights to each of the 6 gift card choices that must all sum up to 100) and is zero otherwise ($n=4,039$). In columns 5 and 6, the dependent variable is a binary variable that takes the value 1 when a participant switches their gift card choice across product categories (for example, from ride-sharing gift cards to a fast food gift card) and is zero otherwise ($n=4,039$). Columns 7 and 8 show regressions for the sub-sample of participants who switch their gift card choice; the dependent variable is the difference in the intensity of advertising misinformation between the participant's top choice gift card company and the company they finally choose after receiving the information treatment ($n=430$). We report summary statistics and demonstrate that our treatment groups are balanced across observable characteristics in Supplementary Table 5. Results for the sample including inattentive respondents are shown in Supplementary Table 9. No adjustments were made for multiple comparisons. Robust standard errors are shown in parentheses. $P$ values derived from two-sided $t$-tests are reported below the standard errors. The remaining pairwise $P$ values are reported in Supplementary Table 6. *$P<0.1$, **$P<0.05$, ***$P<0.01$.

(T1, T3 and T4), where respondents are informed that advertisements from their top choice of gift card company recently appeared on misinformation websites. Table 1, column 1 shows that respondents increasingly exit (that is, increase switching away or decrease demand from) their first choice company relative to control ($b=0.13$, 95% confidence interval [0.10, 0.16], $P<0.001$) in response to learning about their top choice gift card company's advertisements appearing on misinformation websites (T1). This effect persists ($b=0.13$, 95% confidence interval [0.10, 0.16], $P<0.001$; Table 1, column 2) when we control for participants' demographic and behavioural characteristics in our preferred specification, which enables more precise estimates (see Supplementary Information, 'Analysis: consumer study outcomes'). We also use text analysis of the responses to a free-form question, which helps to identify the effect of the information intervention more directly. Respondents' text responses explaining their choice of the gift card reveal that misinformation concerns drive this switching behaviour (Extended Data Fig. 1a).

Switching behaviour also increases relative to the control group ($b=0.10$, 95% confidence interval [0.07, 0.13], $P<0.001$) when respondents are told about the substantial role of digital advertising platforms in placing companies' advertisements on misinformation websites (T3). This switching behaviour persists even though respondents are more likely to state that digital advertising platforms are responsible for placing companies' advertisements on misinformation websites by four percentage points relative to the control group ($b=0.04$, 95% confidence interval [0.02, 0.06], $P<0.001$, Extended Data Fig. 1b). This suggests that advertising companies can continue to experience a decline in demand for their products or services despite consumers knowing that digital advertising platforms have a substantial role in placing companies' advertisements on misinformation websites.

When provided with a ranking of companies in order of their intensity of appearance on misinformation websites (T4), respondents switch away from opting for their top choice gift card company ($b=0.08$, 95%

confidence interval [0.05, 0.11], $P<0.001$). This result shows that the advertising companies can expect to face a decrease in consumption for financing misinformation despite other companies also advertising on misinformation outlets. Respondents are less likely to mention product features that are relevant to the companies they are interested in—for example, healthy food, good prices and availability in the local area, among others ($b=-0.07$, 95% confidence interval [−0.09, −0.05], $P<0.001$, Extended Data Fig. 1a). Examining the direction of consumer switching shows that among those who switch their gift card preference ($n=430$), those provided with company-ranking information in T4 made the most switches towards companies that less frequently advertised on misinformation websites ($b=0.95$, 95% confidence interval [0.19, 1.71], $P=0.015$). This result suggests that providing a ranking of advertising companies transparently could steer consumer demand towards companies that advertise less frequently on misinformation websites.

Our results are robust to alternative exit outcomes that include whether participants switch to a product they prefer less than their first choice (Table 1, columns 3 and 4) and whether they switch their choice across product categories (Table 1, columns 5 and 6), further indicating that participants incur a real cost of switching to a company that is not equivalent to their top-ranked one. Although our platform-related information treatment (T2) does not explicitly mention the respondents' first choice gift company (as in T1, T3 and T4) or its specific use of digital advertising platforms (as in T3), we observe a small amount of switching in T2 relative to the control group ($b=0.03$, 95% confidence interval [0.01, 0.05], $P=0.012$). This could be because respondents might partially blame their first choice gift card company as it could be top of mind for them[42] or assume that the information provided in T2 alluded to the company they had just chosen[43]. It is important to note that the other outcomes reported in Table 1 in the paper—that is, switching to lower preference gift cards and switching across categories are not statistically significant for T2, which suggests that T2 does not

result in treatment effects similar to our other treatments. Overall, we find that companies whose advertisements appear on misinformation websites can face substantial consumer backlash in terms of both exit and voice. Consumers who switched their gift card choice as a result of our information treatments lost about 39.4% of the mean value and 42.9% of the median value of their gift card value on average. Given that the value of the gift card is US$25, a 39.4% decline in the mean value translates to treated consumers losing an equivalent of US$9.85. The distribution of weights assigned to the initial top gift card choice and the final selection is shown in Extended Data Fig. 2, which illustrates a substantial leftward shift in the weight distribution when individuals switch away from their top choice. We also find suggestive evidence for vast differences between consumers' stated and revealed preferences, as shown in Supplementary Fig. 3. When compared to prior research, our 13 percentage point decline in demand is similar in magnitude to the demand reduction observed from receiving negative product feedback[44] and exceeds the magnitude of previously measured changes in demand associated with companies taking a social or political stance[37,38].

Next, we examine the effects of the information interventions on our pre-registered voice outcomes captured by individuals signing an online petition to voice concerns about advertising on misinformation websites. Participants were given the option to sign one of four different petitions on Change.org (https://www.change.org/): two company-level petitions advocating that companies in general should block or allow their advertisements from appearing on misinformation outlets, and two similar platform-level petitions. Although we observe petition signatures at the group level, we use clicks on petition links as our primary voice outcome since this information is available at the individual level and most closely matches the proportions of actual signatures (Extended Data Fig. 3). Our results are robust to using alternative petition outcomes, such as intention to sign a petition, self-reported petition signatures and actual signatures (Extended Data Table 2). Of note, we do not analyse actual signatures for the T4 group since Change.org accidentally deleted these petitions after they were recorded.

Relative to the control group, participants were 5 percentage points (36%) significantly more likely to click on the platform petition link when given information about the role of digital advertising platforms in automatically placing advertisements on misinformation websites in the platform (T2) treatment group (Table 2, columns 3 and 4). Text analysis from respondents' explanation of their petition choice confirms that respondents hold digital advertising platforms more responsible for financing misinformation in T2 relative to the control group ($b = 0.02$, 95% confidence interval [0.01, 0.04], $P = 0.012$, Extended Data Fig. 1b). For example, one respondent stated who opted for the platform blocking petition explained their choice by stating that the platform option "involves more than one company." Another stated that their chosen gift card company is "not the only ad being put on misinformation sites. It is a larger issue that has to do with the platforms used to place ads." Indeed, signing these petitions is the only way that participants can take any action to hold advertising platforms responsible in response to T2, which explicitly highlights the role of platforms.

Upon receiving information about all six gift card companies' advertisements appearing on misinformation websites (T4), participants were significantly more likely to click on petition links suggesting that advertising companies need to block their advertisements from appearing on misinformation websites (Table 2, columns 3 and 4). Based on their open-ended text responses (Extended Data Fig. 1a), respondents increasingly highlighted misinformation-related concerns ($b = 0.09$, 95% confidence interval [0.07, 0.11], $P < 0.001$) and placed less emphasis on product usage ($b = -0.05$, 95% confidence interval [−0.07, − 0.03], $P < 0.001$) and product features ($b = -0.07$, 95% confidence interval [−0.09, − 0.05], $P < 0.001$). In T4, the treatment intensity for companies,

## Table 2 | Average treatment effects on voice

| | Company | | Platform | |
| --- | --- | --- | --- | --- |
| | 1 | 2 | 3 | 4 |
| Company (T1) | 0.02 | 0.02 | −0.02 | −0.02 |
| | (0.02) | (0.02) | (0.02) | (0.02) |
| | 0.180 | 0.257 | 0.286 | 0.349 |
| Platform (T2) | −0.01 | −0.01 | 0.05*** | 0.05*** |
| | (0.02) | (0.02) | (0.02) | (0.02) |
| | 0.546 | 0.007 | 0.436 | 0.006 |
| Company and platform (T3) | −0.00 | −0.00 | −0.01 | −0.01 |
| | (0.02) | (0.02) | (0.02) | (0.02) |
| | 0.863 | 0.940 | 0.638 | 0.708 |
| Company ranking (T4) | 0.04** | 0.04** | −0.03* | −0.03 |
| | (0.02) | (0.02) | (0.02) | (0.02) |
| | 0.047 | 0.049 | 0.080 | 0.115 |
| Controls | No | Yes | No | Yes |
| Control group mean | 0.15 | 0.15 | 0.14 | 0.14 |
| Observations | 4039 | 4039 | 4039 | 4039 |

OLS regression results for each of the four treatment groups (T1, T2, T3 and T4) in our sample ($n = 4,039$). Columns are numbered along the top. In columns 1 and 2, the dependent variable is clicking on a link to sign a petition that suggests that companies like the respondent's top choice gift card company need to block their advertisements from appearing on misinformation websites. In columns 3 and 4, the dependent variable is clicking on a link to sign a petition that suggests that digital advertising platforms used by companies need to block advertisements from appearing on misinformation websites. Alternative petition outcomes are shown in Extended Data Table 2. Alternative petition outcomes are shown in Extended Data Table 2. Results for the sample including inattentive respondents are shown in Supplementary Table 10. No adjustments were made for multiple comparisons. Robust standard errors are shown in parentheses. $P$ values derived from two-sided $t$-tests are reported below the standard errors. The remaining pairwise $P$ values are reported in Supplementary Table 7.

in general, is significantly stronger relative to T1 and T3 since we highlighted that all six gift card companies advertise on misinformation websites (at varying levels). This increase in treatment intensity could explain a higher treatment effect for T4 relative to the null effects for company petitions in the other treatment arms, which only specifically mentioned the respondents' top choice gift card company.

## Heterogeneous treatment effects

Next, we explore heterogeneity in treatment effects along four pre-registered dimensions (gender, political orientation, frequency of use of the company's products or services, and consumption of misinformation) based on our hypotheses (see Methods, 'Consumer experiment design'). Focusing on exit (Extended Data Table 3, columns 1–4), we observe positive treatment effects for all groups — that is, male and female, Biden voters and Trump voters, frequent and infrequent users of a company's products or services, and those who report consuming news from misinformation outlets in our survey and those who do not. As reported in Extended Data Table 3, in line with our predictions, we find stronger treatment effects for exit among women ($b = 0.05$, $P = 0.011$) and Biden voters ($b = 0.03$, $P = 0.058$) and less strong treatment effects for frequent users ($b = -0.05$, $P = 0.007$) and those who consume news from select popular misinformation outlets ($b = -0.04$, $P = 0.097$). Respondents who voted for President Biden in the 2020 US Presidential election were also 5 percentage points more likely to voice concerns against company practices ($P = 0.04$; Extended Data Table 3, column 6). Overall, we believe these heterogeneity results bolster the external validity of our experimental estimates. In particular, we highlight that product-specific factors such as frequency of use can have an important role in the decision to switch or not separately from ideological reasons such as political leaning.

## Measuring decision-maker preferences

Given that advertising on misinformation websites is pervasive and could provoke consumer backlash, we next examine what explains the prevalence of this phenomenon among companies. To shed light on this question, we surveyed key strategic decision-makers such as executives and managers at companies by partnering with the executive education programmes at two universities to survey their alumni. In collaboration with our partner organizations, we also verified the job titles of the majority (71%) of our respondents using external sources, which are shown in Extended Data Fig. 4. About 94% of the participants whose job titles we were able to verify served in a top executive role or managerial role at the time of our survey (for example, chief executive, general or operations manager of multiple departments or locations, advertising or sales manager or operations manager) and the remainder were individuals who could influence decision-making within their companies, especially given their interest in learning leadership and managerial skills via executive education programmes.

## Baseline beliefs and preferences

We found a wide dispersion in decision-makers' pre-registered beliefs about the role of companies and platforms in financing misinformation as shown in Supplementary Fig. 6 and 7, which complements prior work showing wide dispersion in decision-makers' beliefs in other settings[45,46]. Decision-makers largely overestimate the overall proportion of companies that advertise on misinformation websites and underestimate the role of digital advertising platforms in placing companies' advertisements on misinformation websites. In particular, respondents estimated that about 64% of companies' advertisements appeared on misinformation websites on average (Supplementary Table 12). However, our data show that 55% of the 100 most active advertisers appeared on misinformation websites. Regarding the role of digital advertising platforms, respondents estimated that around 44.5% of companies using digital advertising platforms appear on misinformation websites (Supplementary Table 12), whereas 79.8% of companies among the 100 most active advertisers in fact do so. Moreover, only 41% of decision-makers believed that consumers react against companies whose advertisements appear on misinformation websites. These results suggest that decision-makers believe that advertising on misinformation websites is probably commonplace but has little to do with using digital advertising platforms and has limited consequences for the companies involved.

However, in contrast to the average belief that most companies advertised on misinformation websites, respondents substantially underestimated their own company's likelihood of appearing on misinformation websites. Only 20% of respondents believing that their own company's advertisements recently appeared on misinformation websites, which indicates the presence of a false uniqueness effect among decision-makers[47]. We further segmented our results by type of role within the company (Extended Data Table 4). Although our sub-samples were small, these baseline beliefs and characteristics were largely similar across various roles. Among participants who expressed an interest in learning about whether their company's advertisements appeared on misinformation websites (that is, requested an advertisement check by providing their company name and contact details) and whose companies appeared in our advertising data, approximately 81% of companies appeared on misinformation websites. Moreover, most respondents who were given follow-up information that their companies' advertisements appeared on misinformation websites reported being surprised by this information (62%), whereas none of those who learned their companies advertisements did not appear on misinformation websites reported being surprised. These figures illustrate that decision-makers are largely uninformed about the high likelihood of their company's advertisements appearing on misinformation

### Table 3 | Average treatment effects of information intervention

| | Posterior platform belief | | | Platform solution demand | | |
|---|---|---|---|---|---|---|
| | **All** | **Yes** | **No** | **All** | **Yes** | **No** |
| | **1** | **2** | **3** | **4** | **5** | **6** |
| Treatment | 48.99*** | 6.11 | 54.15*** | −0.03 | −0.10 | −0.03 |
| | (15.79) | (43.43) | (17.77) | (0.05) | (0.13) | (0.05) |
| | 0.002 | 0.883 | 0.002 | 0.559 | 0.436 | 0.567 |
| Controls | Yes | Yes | Yes | Yes | Yes | Yes |
| Control group mean | 96.10 | 124.47 | 88.65 | 0.37 | 0.34 | 0.38 |
| Observations | 442 | 88 | 354 | 442 | 88 | 354 |

OLS regression results where the dependent variables are posterior beliefs (columns 1–3) and demand for platform solution (columns 4–6). Columns are numbered along the top. We winsorize the posterior beliefs to remove outliers. Our platform solution demand outcome variable is a binary variable that takes a value of one when participants choose to receive information on which platforms least frequently place companies' advertisements on misinformation websites and zero otherwise. Columns 1 and 4 show results for the full sample of participants (n = 442). Columns 2 and 5 show results for the sub-sample of participants who reported 'yes' to the question "Do you think your company or organization had its ads appear on misinformation websites during the past three years (2019–2021)?" (n = 88). Columns 3 and 6 show results for the sub-sample who reported 'No' in response to the same question (n = 354). We control for decision-makers' characteristics and prior beliefs. Results for the sample including inattentive respondents are shown in Supplementary Table 14. No adjustments were made for multiple comparisons. Robust standard errors are shown in parentheses. P values derived from two-sided t-tests are reported below the standard errors.

websites. Given these findings about the beliefs of decision-makers, our results suggest that companies may be financing misinformation inadvertently.

Most participants requested an advertisement check by providing their company name and email address (74%). The demand for an advertisement check was high regardless of respondents' initial beliefs, suggesting a substantial interest in learning about whether their company's advertisements appeared on misinformation websites. Despite only 41% of respondents agreeing that consumers react against companies whose advertisements appear on misinformation websites, most participants (73%) opted to receive information on how consumers respond to companies whose advertisements appear on misinformation websites with 58% inquiring about exit and 15% enquiring about voice. This suggests that although decision-makers may be unaware of how advertising on misinformation websites can provoke consumer backlash, most of them are interested in learning about the degree of potential backlash. Finally, for our most costly revealed-preference measure—that is, signing up to attend a 15-minute expert-led information session on how companies can avoid advertising on misinformation websites—18% of decision-makers opted to sign up, an arguably high rate given the value of decision-makers' time and the opportunity cost of attending the session.

## Information intervention results

We report the results of our information treatment on our pre-registered outcomes. For the full sample of participants, we estimate positive and statistically significant effects on participants' posterior beliefs about the role of advertising platforms in placing advertisements on misinformation websites (Table 3, column 1), driven mainly by respondents who believe that their company's advertisements had not appeared on misinformation websites in the recent past (Table 3, column 3).

We find an overall null effect of our information treatment on participants' demand for a platform-based solution, as measured by their demand for information on which platforms least frequently place companies' advertisements on misinformation websites

**Table 4 | Treatment effects based on prior beliefs**

| | Posterior platform belief | | Platform solution demand | |
|---|---|---|---|---|
| | Certain | Uncertain | Certain | Uncertain |
| | **1** | **2** | **3** | **4** |
| Treatment | 39.98** | 144.25** | −0.07 | 0.36*** |
| | (20.23) | (60.23) | (0.06) | (0.13) |
| | 0.049 | 0.022 | 0.213 | 0.008 |
| Controls | Yes | Yes | Yes | Yes |
| Control group mean | 90.23 | 80.80 | 0.37 | 0.43 |
| Observations | 286 | 68 | 286 | 68 |

OLS regression results for the sub-sample of participants who reported 'No' to the question "Do you think your company or organization had its advertisements appear on misinformation websites during the past three years (2019–2021)?" (n = 354). Columns are numbered along the top. The dependent variables are posterior beliefs (columns 1 and 2) and demand for platform solution (columns 3 and 4) from Table 3. Columns 1 and 3 show results for participants who report being certain about their response to the question—that is, choosing 'Somewhat sure', 'Sure' or 'Very sure' (n = 286). Column 4 shows results for participants who report being uncertain about their response to the question—that is, choosing 'Unsure' or 'Very unsure' (n = 68). Results for the sample including inattentive respondents are shown in Supplementary Table 15. No adjustments were made for multiple comparisons. Robust standard errors are shown in parentheses. P values derived from two-sided t-tests are reported below the standard errors.

(Table 3, columns 4–6). However, this result masks substantial heterogeneity based on participants' prior beliefs. Since our information treatment changes beliefs for the subset of participants who believe that their company's advertisements had not recently appeared on misinformation websites (Table 3, column 3), we further investigated and reported results based on participant's prior beliefs for this sub-sample in Table 4. Only participants who were uncertain and unaware about their own company's advertisements appearing on misinformation websites responded positively and significantly to our information treatment by increasing their demand for a platform-based solution by 36 percentage points (b = 0.36, 95% confidence interval [0.11, 0.61], P = 0.008, n = 68), as shown in Table 4, column 4. Our results imply that the way in which participants respond to information about the role of digital advertising platforms in financing misinformation is highly dependent on their prior beliefs about their own company. Such information could make companies switch advertising platforms or pressure the platforms they currently use to enable them to easily steer their advertising away from misinformation outlets. This finding is in line with a lack of attention describing decision-makers' behaviours across various settings[48–50]. However, these results should be viewed as suggestive and exploratory since the subsample sizes in these regressions are small and these sample splits were not pre-registered.

We did not find meaningful treatment effects for our donation preference outcome, which measures the proportion of respondents who prefer that we donate to the GDI instead of DataKind (Supplementary Table 13). Since both GDI and DataKind have similar goals of advancing technology's ethical and responsible use, respondents may have considered their missions interchangeable. Moreover, unlike our first behavioural outcome, respondents could have considered donating to the GDI less relevant to their own organizations' needs and more a matter of personal preference.

## Discussion

Together, our descriptive and experimental findings offer clear, practical implications. Given the potential for a substantial decline in demand, as demonstrated by our consumer study, advertising companies may wish to account for consumer preferences in placing their advertising across various online outlets and exercise caution while incorporating automation in their business processes via digital advertising platforms. For instance, given that consumers switched to other products upon learning about a company's advertisements appearing on misinformation websites, companies could use lists of misinformation outlets provided by independent third-party organizations such as NewsGuard and the GDI to limit advertising budgets being spent on misinformation outlets through digital platforms. Moreover, since consumer backlash was particularly strong for women and politically left-leaning consumers, companies targeting such audiences may need to exercise greater caution.

On the basis of our results, we identify two interventions that could reduce the financing of online misinformation. First, digital advertising platforms that run automated auctions could enable advertisers to more easily access data on whether their advertisements appear on misinformation outlets. This would enable advertisers to make advertising placement decisions consistent with their preferences rather than inadvertently financing misinformation[51]. Second, while it is currently possible for consumers to find out about companies financing misinformation through media reports, digital platforms could improve transparency for consumers about which companies advertise on misinformation outlets. Platforms could provide such information to consumers when they are viewing an advertisement using simple information labels (as in our 'company only' information treatment) similar to the 'sponsored by' and 'paid for by' labels that are presently common on various digital media platforms. Similarly, rank-based information provided in our company-ranking information treatment (T4) could be provided as a ranking of companies in order of intensity of appearing on misinformation websites where customers are selecting products from a menu of choices while shopping. Platforms have provided similar contextual information about companies in other settings—for example, Google Flights displays carbon emissions data alongside flight prices when people select a flight to purchase among several options[52]. Enabling consumers to view such information at the point of purchase could provide a stronger incentive for companies to steer their advertisements away from such outlets, especially since the effect of negative information can persist for several months[53]. Overall, these interventions could decrease the inadvertent advertising revenue going towards misinformation outlets, which could eventually lead to such sites ceasing to operate, as observed anecdotally in prior work[29].

These interventions could ensure that both consumers and advertisers are provided with information about the consequences of their respective purchasing and advertising placement decisions so that they can account for their preferences. Having access to such information is necessary for an efficiently functioning economic system in accordance with the first fundamental theorem of welfare economics. However, whereas digital platforms are uniquely well-positioned in the ecosystem of consumers, advertisers and publishers to implement information interventions in the form of disclosures and rankings[54,55], they may not have incentives to implement such interventions. With the backdrop of mounting pressure from advertisers[27,28] and calls for transparency in the programmatic advertising business[56], information-based interventions could be incorporated into existing legislation to improve transparency. These include efforts such as the EU Digital Services Act, which includes a Code of Practice on Disinformation with enforceable provisions for different stakeholders in the advertising ecosystem to collectively fight misinformation, and US bills such as the Honest Ads Act and the Competition and Transparency in Digital Advertising (CTDA) Act, which include provisions to improve transparency in political advertising and the digital advertising ecosystem in general. Notably, in recent years, policy proposals that aim to reduce the prevalence of misinformation such as the Combating Misinformation and Disinformation bill in Australia and the bill against fake news in Germany have faced backlashes over posing risks to free speech[57,58]. Although such proposals face the challenge of striking the right balance between combating misinformation and protecting freedom of

expression, the information interventions that we identify could help counter the financial incentive to produce misinformation in the first place by reducing the unintended advertising revenue going towards misinformation outlets. There are many parallels for regulation by information provision to address externalities in other industries, including chemicals (toxic release inventory reporting requirements), automobiles (fuel consumption information), food (nutrition and content labels) and airlines (greenhouse gas emissions), of which several have been demonstrated to be effective in prior work[41,59,60].

Previously studies have shown that 'demand-side' interventions to counter online misinformation have focused on reducing the consumption and spread of misinformation among news consumers on online platforms. Although interventions such as accuracy prompts and digital literacy tips can increase the quality of news that people share[5], this line of work has found limited support for news credibility signals in increasing the demand for credible news[61] or in reducing misperceptions among users[6]. Such constraints in changing user behaviour may also apply to credibility signals like watermarks for detecting AI misinformation. Moreover, whereas such interventions are only effective for the small subset of users who are exposed to misinformation[62], our complementary 'supply-side' approach targets entities and individuals who might not necessarily consume or spread misinformation themselves.

Relative to existing proposals of supply-side interventions to curb the production of misinformation, which involve social media platforms banning the advertising of false news[63] or changing their advertising-driven business model altogether[18], we outline a middle path to suggest that accounting for the preferences of advertisers and consumers could help counter the financing of online misinformation. Although platforms could coordinate to identify and deplatform misinformation websites[64], prior work suggests that misinformation websites nearly always resurface through alternative providers unless the incentive to produce misinformation is addressed[29]. Moreover, the information interventions that we identify are also an improvement on the status quo, whereby advertisers and consumers can only implement their preferences by participating in boycotts of digital platforms over their inability to contain misinformation. Allowing advertisers to more easily observe and control whether their advertisements appear on misinformation websites could also limit backlash by enabling advertisers to better implement their preferences rather than participating in one-off short-term advertising boycotts[27,28]. Additionally, since consistently providing negative information can create lasting associations for consumers[65], providing information disclosures on every advertisement for whether the advertising company involved appears on misinformation websites could have a substantial effect on consumer demand over time, providing incentives for advertising companies to reduce advertising on misinformation websites.

Given our findings, we suggest three promising avenues for future research. First, future work could evaluate the effectiveness of our information interventions in the field over a longer time period to quantify the decline in revenue generated by misinformation outlets resulting from increasing transparency for consumers or advertisers. Related to this, future work could also target a wider set of advertisers to validate the robustness of our interventions which would allow for broader generalizability. Second, our results on whether companies are willing to adopt solutions to avoid monetizing misinformation are based on their existing (often incorrect) beliefs about the prevalence of advertising on misinformation websites in general and for their own company. More research is needed to understand how advertising companies would respond in the context of correct beliefs. Third, although our research identifies potential interventions that digital platforms can adopt to curb the monetization of online misinformation, it is unclear whether it is in the interest of digital advertising platforms to do so. Moreover, whether the potential monetary and societal benefits of the information interventions we identify outweigh the revenue platforms generate by serving advertisements on misinformation websites remains to be studied. Overall, the effectiveness of platforms in mitigating misinformation will depend on a multi-pronged approach. Given that misinformation is largely financially motivated and that financially sustaining online misinformation can be substantially harmful for the advertising companies involved, simple low-cost informational interventions such as the ones we identify could go a long way in curbing the supply of online misinformation.

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

## Methods

### Background on digital advertising

The predominant business model of several mainstream digital media platforms relies on monetizing attention via advertising[3]. While these platforms typically offer free content and services to individual consumers, they generate revenue by serving as an intermediary or advertising exchange connecting advertisers with independent websites that want to host advertisements. To do so, platforms run online auctions to algorithmically distribute advertising across websites, known as 'programmatic advertising'. For example, Google distributes advertising in this manner to more than two million non-Google sites that are part of the Google Display Network. This allows websites to generate revenue for hosting advertising, and they share a percentage of this payment with the platform. In the USA, more than 80% of digital display advertisements are placed programmatically[16]. We refer to these advertising exchanges as digital advertising platforms and use the term digital platforms to collectively refer to all the services offered by such media platforms.

We examine the role of advertising companies and digital advertising platforms in monetizing online misinformation. While in other forms of (offline) media, advertisers typically have substantial control over where their advertisements appear, advertising placement through digital advertising platforms is mainly automated. Since most companies do not have the capacity to participate in high-frequency advertising auctions that require them to place individual bids for each advertising slot they are interested in, they typically outsource the bidding process to an advertising platform. Such programmatic advertising gives companies relatively less control over where their advertisements end up online. However, companies can take steps to reduce advertising on misinformation websites, such as by only being part of advertising auctions for a select list of credible websites or blocking advertisements from appearing on specific misinformation outlets.

### Collecting website data

We collect data on misinformation websites in three steps. First, we use a dataset maintained by NewsGuard. This company rates all the news and information websites that account for 95% of online engagement in each of the five countries where it operates. Journalists and experienced editors manually generate these ratings by reviewing news and information websites according to nine apolitical journalistic criteria. Recent research has used this dataset to identify misinformation websites[6,66,67]. In this paper, we consider each website that NewsGuard rates as repeatedly publishing false content between 2019 and 2021 to be a misinformation website and all others to be non-misinformation websites, leading to a set of 1,546 misinformation websites and 6,499 non-misinformation websites. To get coverage throughout our study period, we sample websites provided by NewsGuard from the start, middle and end of each year from 2019 to 2021. Additionally, we also sample websites from January 2022 and June 2022 to account for websites that may have existed during our study period and discovered later. Supplementary Table 3 summarizes the characteristics of this dataset. Our NewsGuard dataset contains websites across the political spectrum, including left-leaning websites (for example, https://www.palmerreport.com/ and https://occupydemocrats.com/), politically neutral websites (for example, https://rt.com/ and https://www.nationalenquirer.com/), and right-leaning websites (for example, https://www.thegatewaypundit.com/ and http://theconservativetreehouse.com/).

Note that prior research that has used the NewsGuard dataset has often used the term 'untrustworthy' to describe websites[6,67]. Such research has used NewsGuard's aggregate classification whereby a site that scores below a certain threshold (60 points) on NewsGuard's weighted score system is labelled as untrustworthy. Instead of using NewsGuard's overall score for a website, we use the first criterion classified by NewsGuard for each website—that is, whether a website repeatedly publishes false news to identify a set of 1,546 misinformation websites. While 94% of the NewsGuard misinformation websites we identify in this manner are also untrustworthy based on NewsGuard's classification, only about 52% of the untrustworthy websites are misinformation websites or websites that repeatedly publish false news. Our measure of misinformation is, therefore, more conservative than prior work using NewsGuard's 'untrustworthy' label.

In addition to the NewsGuard dataset, we use a list of websites provided by the GDI. This non-profit organization identifies disinformation by analysing both the content and context of a message, and how they are spread through networks and across platforms[68]. In this way, GDI maintains a list of monthly-updated websites, which it also shares with interested advertising tech platforms to help reduce advertising on misinformation websites. The GDI list allows us to identify 1,869 additional misinformation websites. Finally, we augment our list of misinformation websites with 396 additional ones used in prior work[69,70]. Among the websites that NewsGuard rated as non-misinformation (at any point in our sample), 310 websites were considered to be misinformation websites by our other sources or by NewsGuard itself (during a different period in our sample). We categorize these websites as misinformation websites given their risk of producing misinformation.

Altogether, our website dataset consists of 10,310 websites, including 3,811 misinformation and 6,499 non-misinformation websites. Similar to prior work[6,67], our final measure of misinformation is at the level of the website or online news outlet. Aggregating article-level information and using website-level metadata is meaningful since it reduces noise when arriving at a website-level measure. Finally, we use data from SEMRush, a leading online analytics platform, to determine the level of monthly traffic received by each website from 2019 to 2021.

### Consumer experiment design

This study was reviewed by the Stanford University Institutional Review Board (Protocol No. IRB-63897) and the Carnegie Mellon University Institutional Review Board (protocol no. IRB00000603). Our study was pre-registered at the American Economic Association's Registry under AEARCTR-0009973. Informed consent was obtained from all participants at the beginning of the survey.

**Setting and sample recruitment.** We recruited a sample of US internet users via CloudResearch. CloudResearch screened respondents for our study so that they are representative of the US population in terms of age, gender and race based on the US Census (2020). It is important to note that while we recruited our sample to be representative on these dimensions to improve the generalizability and external validity of our results, our sample is a diverse sample of US internet users, which is not necessarily representative of the US population on other dimensions[71]. To ensure data quality, we include a screener in our survey to check whether participants pay attention to the information provided. Only participants who pass this screener can proceed with the survey. Our total sample includes 4,039 participants, who are randomized into five groups approximately evenly.

The flow of the survey study is shown in Supplementary Fig. 1. We begin by asking participants to report demographics such as age, gender and residence. From a list of trustworthy and misinformation outlets, we then ask participants questions about their behaviours in terms of the news outlets they have used in the past 12 months, their trust in the media (on a 5-point scale), the online services or platforms they have used and the number of petitions they have signed in the past 12 months.

**Initial gift card preferences.** We then inform participants that one in five (that is, 20% of all respondents) who complete the survey will be offered a US$25 gift card from a company of their choice out of six company options. Respondents are asked to rank the six gift card companies on a scale from their first choice (most preferred) to their

sixth choice (least preferred). These six companies belong to one of three categories: fast food, food delivery and ride-sharing. All six companies appeared on the misinformation websites in our sample during the past three years (2019–2021), offer items below US$25, and are commonly used throughout the USA. The order in which the six companies are presented is randomized at the respondent level. As a robustness check, we also ask respondents to assign weights to each of the six gift card options. This question gives respondents greater flexibility by allowing them to indicate the possibility of indifference (that is, equal weights) between any set of options. We then ask participants to confirm which gift card they would like to receive if they are selected to ensure they have consistent preferences regardless of how the question is asked. At this initial elicitation stage, the respondents did not know that they will get another chance to revise their choice. Hence, these choices can be thought of as capturing their revealed preference.

**Information treatments.** All participants in the experiment are given baseline information on misinformation and advertising. This is meant to ensure that all participants in our experiment are made aware of how we define misinformation along with examples of a few misinformation websites (including right-wing, neutral and left-wing misinformation websites), how misinformation websites are identified, and how companies advertise on misinformation websites (via an illustrative example) and use digital platforms to automate placing advertisements.

Participants are then randomized into one control and four treatment groups, in which the information treatments are all based on factual information from our data and prior research. We use an active control design to isolate the effect of providing information relevant to the practice of specific companies on people's behaviour[9]. Participants in the control group are given generic information based on prior research that is unrelated to advertising companies or platforms but relevant to topic of news and misinformation.

In our first 'company only' treatment group (T1), participants are given factual information stating that advertisements from their top choice gift card company appeared on misinformation websites in the recent past. Based on their preferences, people may change their final gift card preference away from their initial top-ranked company after receiving this information. It is unclear, however, whether advertising on misinformation websites would cause a sufficient change in consumption patterns and which sets of participants may be more affected.

Our second 'platform only' treatment group (T2) informs participants that companies using digital advertising platforms were about 10 times more likely to appear on misinformation websites than companies that did not use such platforms in the recent past. This information treatment measures the effects of digital advertising platforms in financing misinformation news outlets. Since it does not contain information about advertising companies, it practically serves as a second control group for our company-level outcome and aims to measure how people may respond to our platform-related outcome.

Because our descriptive data suggest that the use of digital advertising platforms amplifies advertising revenue for misinformation outlets, we are interested in measuring how consumers respond to a specific advertising company appearing on misinformation websites when also informed of the potential role of digital advertising platforms in placing companies' advertising on misinformation websites. It is unclear whether consumers will attribute more blame to companies or advertising platforms for financing misinformation websites when informed about the role of the different stakeholders in this ecosystem. For this reason, our third 'company and platform' treatment (T3) combines information from our first two treatments (T1 and T2). Similar to T1, participants are given factual information that advertisements from their top choice gift card company appeared on misinformation websites in the recent past. Additionally, we informed participants that their top choice company used digital advertising platforms and

companies that used such platforms were about ten times more likely to appear on misinformation websites than companies that did not use digital advertising platforms, as mentioned in T2.

Finally, since several advertising companies appear on misinformation websites, we would like to determine whether informing consumers about other advertising companies also appearing on misinformation websites changes their response towards their top choice company. In our fourth company-ranking treatment (T4), participants are given factual information, which states that "In the recent past, ads from all six companies below repeatedly appeared on misinformation websites in the following order of intensity", and provided with a ranking from one of three years in our study period—that is, 2019, 2020 or 2021. We personalize these rankings by providing truthful information based on data from different years in the recent past such that the respondents' top gift card choice company does not appear last in the ranking (that is, is not the company that advertises least on misinformation websites) and in most cases, advertises more intensely on misinformation websites than its potential substitute in the same company category (for example, fast food, food delivery or ride-sharing). Such a treatment allows us to measure potential differences in the direction of consumers switching their gift card choices, such as switching towards companies that advertise more or less intensely on misinformation websites. It could also give consumers reasonable deniability such as "everyone advertises on misinformation websites" leading to ambiguous predictions about the exact impact of the treatment effect.

**Outcomes.** We measure two pre-registered behavioural outcomes that collectively allow us to measure how people respond to our information treatments in terms of both voice and exit[25]. After the information treatment, all participants are asked to make their final gift card choice from the same six options they were shown earlier. Our main outcome of interest is whether participants 'exit' or switch their gift card preference—that is, whether they select a different gift card after the information treatment than their top choice indicated before the information treatment. To ensure incentive compatibility, participants are (truthfully) told that those randomly selected to receive a gift card will be offered the gift card of their choice at the end of our study. As mentioned above, the probability of being randomly chosen to receive a gift card is 20%. We choose a high probability of receiving a gift card relative to other online experiments since prior work has shown that consumers process choice-relevant information more carefully as realization probability increases[72]. To make the gift card outcome as realistic as possible, we also had a large value gift card (US$25). The focus of our experiments is on single-shot outcomes. While it would have been interesting to capture longer-term effects, the cost of implementing our gift card outcome for a large sample and expenditure on the other studies made a follow-up study cost-prohibitive.

Secondly, participants are given the option to sign one of several real online petitions that we made and hosted on Change.org. Participants can opt to sign a petition that advocates for either blocking or allowing advertising on misinformation or choose not to sign any petition. Further, participants could choose between two petitions for blocking advertisements on misinformation websites, suggesting that either: (1) advertising companies, or (2) digital advertising platforms, need to block advertisements from appearing on misinformation websites. Overall, participants selected among the following five choices: (1) "Companies like X need to block their ads from appearing on misinformation websites.", where X is their top choice gift card company; (2) "Companies like X need to allow their ads to appear on misinformation websites.", where X is their top choice gift card company; (3) "Digital ad platforms used by companies need to block ads from appearing on misinformation websites."; (4) "Digital ad platforms used by companies need to allow ads to appear on misinformation websites."; and (5) I do not want to sign any petition. To track the number of petition signatures for each of these four petition options across our randomized groups,

we provide separate petition links to participants in each randomized group. We record several petition-related outcomes. First, we measure participants' intention to sign a petition based on the option they select in this question. Participants who pass our attention check and opt to sign a petition are later provided with a link to their petition of choice. This allows tracking whether participants click on the petition link provided. Participants can also self-report whether they signed the petition. Finally, for each randomized group, we can track the total number of actual petition signatures.

Our petition outcomes serves two purposes. While our gift card outcome measures how people change their consumption behaviour in response to the information provided, people may also respond to our information treatments in alternative ways—for example, by voicing their concerns or supplying information to the parties involved[25,26]. Given that the process of signing a petition is costly, participants' responses to this outcome would constitute a meaningful measure similar to petition measures used in prior experimental work[73,74]. Second, since participants must choose between signing either company or platform petitions, this outcome allows us to measure whether or not, across our treatments, people hold advertising companies more responsible for financing misinformation than the digital advertising platforms that automatically place advertisements for companies.

In addition to our behavioural outcomes, we also record participants' stated preferences. To do so, we ask participants about their degree of agreement with several statements about misinformation on a seven-point scale ranging from 'strongly agree' to 'strongly disagree'. These include whether they think: (1) companies have an important role in reducing the spread of misinformation through their advertising practices; and whether (2) digital platforms should give companies the option to avoid advertising on misinformation websites.

**Heterogeneous treatment effects.** We explore heterogeneity in consumer responses along four pre-registered dimensions. First, prior research recognizes differences in the salience of prosocial motivations across gender[75], with women being more affected by social-impact messages than men[76] and more critical consumers of new media content[77]. Given these findings, we could expect female participants to be more strongly affected by our information treatments.

Responses to our information treatments may also differ by respondents' political orientation. According to prior research, conservatives are especially likely to associate the mainstream media with the term 'fake news'. These perceptions are generally linked to lower trust in media, voting for Trump, and higher belief in conspiracy theories[78]. Moreover, conservatives are more likely to consume misinformation[2] and the supply of misinformation has been found to be higher on the ideological right than on the left[79]. Consequently, we might expect stronger treatment effects for left-wing respondents.

Consumers who more frequently use a company's products or services could be presumed to be more loyal towards the company or derive greater utility from its use, which could limit changes in their behaviour[37]. Alternatively, more frequent consumers may be more strongly affected by our information treatments as they may perceive their usage as supporting such company practices to a greater extent than less frequent consumers.

Finally, we measure whether people's responses differ by whether they consume misinformation themselves based on whether they reported using misinformation outlets in the initial question asking them to select which news outlets they used in the past 12 months.

**Tackling experimental validity concerns.** In our incentivized, online setting where we measure behavioural outcomes, we expect experimenter demand effects to be minimal as has been evidenced in the experimental literature[80,81]. We take several steps to mitigate potential experimenter demand effects, including implementing best practices recommended in prior work[9]. First, our experiment has a neutral framing throughout the survey since the recruitment of participants. While recruiting participants, we invite them to "take a survey about the news, technology and businesses" without making any specific references to misinformation or its effects. While introducing misinformation websites and how they are identified by independent non-partisan organizations, we include examples of misinformation websites across the political spectrum (including both right-wing and left-wing sites) and provide an illustrative example of misinformation by foreign actors. In drafting the survey instruments, the phrasing of the questions and choices available were as neutral as possible. For example, while introducing our online petitions, we presented participants with the option to sign real petitions that suggest both blocking and allowing advertising on misinformation sites. Indeed, we find that the vast majority of participants believe that the information provided in the survey was unbiased as shown in Supplementary Fig. 4. Only about 10% of participants chose one of the 'biased' or 'very biased' options when asked to rate the political bias of the survey information provided from a seven-point scale ranging from 'very right-wing biased' to 'very left-wing biased'.

In our active control design, participants in all randomized groups are presented with the same baseline information about misinformation, given misinformation-related information in the information intervention and asked the same questions after the information intervention to emphasize the same topics and minimize potential differences in the understanding of the study across treatment groups. Moreover, to maximize privacy and increase truthful reporting[82], respondents complete the surveys on their own devices without the physical presence of a researcher. We also do not collect respondents' names or contact details (with the exception of eliciting emails to provide gift cards to participants at the end of the study).

In presenting our information interventions and measuring our behavioural outcomes, we take special care to not highlight the names of the specific entities being randomized across groups to avoid emphasizing what is being measured. We do, however, highlight our gift card incentives by putting the gift card information in bold text to ensure incentive compatibility since prior work has found that failing to make incentives conspicuous can vastly undermine their ability to shift behaviour[83].

Apart from making the above design choices to minimize experimenter demand effects, we measure their relevance using a survey question. Since demand effects are less likely a concern if participants cannot identify the intent of the study[9], we ask participants an open-ended question—that is, "What do you think is the purpose of our study?". Following prior work[84,85], we then analyse the responses to this question to examine whether they differ across treatment groups. To measure potential differences in the respondents' perceptions of the study, we examine their open-ended text responses about the purpose of the study using a Support Vector Machine classifier, which incorporates several features in text analysis, including word, character, and sentence counts, sentiments, topics (using Gensim) and word embeddings. We predict treatment status using the classifier, keeping 75% of the sample for the training set and the remaining 25% as the test set. The classifier predicts treatment status similar to chance for our main treatment groups relative to the control group, as shown in Supplementary Table 11. These results, which are similar in magnitude to those found in previous research[84,85], suggest that our treatments do not substantially affect participants' perceptions about the purpose of the study. Overall, this analysis gives us confidence that our main experimental findings are unlikely to be driven by experimenter demand effects.

To address external validity concerns, we incorporate additional exit outcomes in the paper, showing that treated individuals switched to lower preference products (Table 1, columns 3 and 4) and products across categories (Table 1, columns 5 and 6) after our information interventions by 8 and 5 percentage points, respectively. We also show in Supplementary Table 8 that as the difference between

participants' highest weighted and second highest weighted gift card choice increases, their switching behaviour decreases. This shows that the weights assigned by participants to their gift card options are capturing meaningful and costly differences in value, highlighting the external validity of our findings. More generally, our pre-registered heterogeneity analysis lends credence to the study's external validity. In line with expectations, we find that less frequent users and more politically liberal individuals are likelier to switch (see Extended Data Table 3 for the full set of pre-registered heterogeneity results). Moreover, we find that the cost of switching gift cards varies based on participants' observable characteristics. For example, treated participants who reported not using any of the misinformation news outlets in our survey lost 50% of the median value (US$12.50) of their initial top choice gift card whereas treated participants who reported reading such outlets lost 33.3% of the median value (US$8.33) of their initial top choice gift card. Participants' text responses also indicate that they believed their choices to be consequential (see Supplementary Tables 1 and 2). As an example, while explaining their choice of gift card, one participant stated, "Because I would most likely use this gift card on my next visit to… and it is less likely that i would use the others." Regarding the petition outcome, one participant stated "The source of this problem seems to be from the digital advertising platforms, so I'd rather sign the petition that stops them from putting ads on misinformation websites."

### Decision-maker experiment design

We followed the same IRB review, pre-registration and consent procedures as those used for our consumer study. This study addresses two research questions. First, we aim to measure the existing beliefs and preferences decision-makers have about advertising on misinformation websites. This will help inform whether companies may be inadvertently or willingly sustaining online misinformation. Secondly, we ask: how do decision-makers update their beliefs and demand for a platform-based solution to avoid advertising on misinformation websites in response to information about the role of platforms in amplifying the financing of misinformation? This will suggest whether companies may be more interested in adopting advertising platforms that reduce the financing of misinformation. To this end, we conduct an information-provision experiment[9]. While past work has examined how firm behaviour regarding market decisions changes in response to new information[48,49], it is unclear how information on the role of digital advertising platforms in amplifying advertising on misinformation would affect decision-makers' non-market strategies.

**Setting and sample recruitment.** To recruit participants, we partnered with the executive education programmes at the Stanford Graduate School of Business and Heinz College at Carnegie Mellon University. We did so in order to survey senior managers and leaders who could influence strategic decision-making within their firms, in contrast to studies relying heavily on MBA students for understanding decision-making in various contexts such as competition, pricing, strategic alliances and marketing[86–89]. Additionally, partnering with two university programmes instead of a specific firm allowed us to access a more diverse sample of companies than prior work that sampled specific types of firms—for example, innovative firms, startups or small businesses[90–92]. Throughout this study, we use the preferences of decision-makers (for example, chief executive officers) as a proxy for company-level preferences since people in such roles shape the outcomes of their companies through their strategic decisions[93,94].

Our partner organizations sent emails to their alumni on our behalf. We used neutral language in our study recruitment emails to attract a broad audience of participants to our survey regardless of their initial beliefs and concerns about misinformation, stating our goal as "conducting vital research on the role of digital technologies in impacting your organization" without mentioning misinformation. We received

567 complete responses, of which 90% are kept since they are from currently employed respondents. To ensure data quality, we dropped an additional 13% of responses where participants were inattentive in answering the survey, resulting in a final sample of 442 responses. These participants were determined to be inattentive since they provided an answer greater than 100 when asked to estimate a number out of 100 in the two questions eliciting their prior beliefs about companies and platforms before the information treatment was provided. Our final sample of 442 respondents is from companies that span all the 23 industries in our descriptive analysis. Moreover, as shown in Supplementary Fig. 5, our sample of participants represents a broad array of company sizes and experience levels at their current roles. Additionally, about 22% of the executives in our sample (and 25% of all our participants) are women, which is aligned with the 21% to 26% industry estimates of women in senior roles globally[95,96].

Supplementary Fig. 2 shows the design of the survey study. We first elicit participants' current employment status. All those working in some capacity are allowed to continue the survey, whereas the rest of the participants are screened out. After asking for their main occupation, all participants in the experiment are provided with baseline information on misinformation and advertising similar to that provided in the consumer experiment.

**Baseline beliefs and preferences.** In our pre-registration, we highlighted that we would measure the baseline beliefs and preferences of decision-makers. We measure participants' baseline beliefs about the roles of companies in general, their own company and platforms in general in financing misinformation. Specifically, participants are asked to estimate the number of companies among the most active 100 advertisers whose advertisements appeared on misinformation websites during the past three years (2019–2021). Additionally, we ask participants to report whether they think their company or organization had its advertisements appear on misinformation websites in the past three years. Finally, we measure participants' beliefs about the role of digital advertising platforms in placing advertisements on misinformation websites. To do so, we first inform participants that during the past three years (2019–2021), out of every 100 companies that did not use digital advertising platforms, eight companies appeared on misinformation websites on average. We then asked participants to provide their best estimate for the number of companies whose advertisements appeared on misinformation websites out of every 100 companies that did use digital advertising platforms.

In addition to recording participants' stated preferences using self-reported survey measures, we measure participants' revealed preferences. To ensure incentive compatibility, participants are asked three questions in a randomized order: (1) information demand about consumer responses—that is, whether they would like to learn how consumers respond to companies whose advertisements appear on misinformation websites (based on our consumer survey experiment); (2) advertisement check—that is, whether they would like to know about their own company's advertisements appearing on misinformation websites in the recent past; and (3) demand for a solution—that is, whether they would like to sign up for a 15-minute information session on how companies can manage where their advertisements appear online. Participants are told they can receive information about consumer responses at the end of the study if they opt to receive it whereas the advertisement check and solution information are provided as a follow-up after the survey. Participants are required to provide their emails and company name for the advertisement check. To sign up for an information session from our industry partner on a potential solution to avoid advertising on misinformation websites, participants sign up on a separate form by providing their emails. Since all three types of information offered are novel and otherwise costly to obtain, we expect respondents' demand for such information to capture their revealed preferences.

**Information intervention.** Participants are then randomized into a treatment group, which receives information about the role of digital advertising platforms in placing advertising on misinformation websites, and a control group, which does not receive this information. Based on the dataset we assembled, participants are given factual information that companies that used digital advertising platforms were about ten times more likely to appear on misinformation websites than companies that did not use such platforms in the recent past. This information is identical to the information provided to participants in the T2 (that is, platform only) group in the consumer experiment.

**Outcomes.** After the information intervention, we first measure participants' posterior beliefs about the role of digital advertising platforms in placing advertisements on misinformation websites following our pre-registration. Participants are told about the average number of companies whose advertisements appear per month on misinformation websites that are not monetized by digital advertising platforms. They are then asked to estimate the average number of companies whose advertisements appear monthly on misinformation websites that use digital advertising platforms. This question measures whether participants believe that the use of digital advertising platforms amplifies advertising on misinformation websites.

We record two behavioural outcomes, which were pre-registered as our primary outcomes of interest after the information intervention. Our main outcome of interest is the respondents' demand for a platform-based solution to avoid advertising on misinformation websites. Participants can opt to learn more about two different types of information—that is: (1) which platforms least frequently place companies' advertising on misinformation websites; and (2) which types of analytics technologies are used to improve advertising performance—or opt not to receive any information. Since participants can only opt to receive one of the two types of information, this question is meant to capture the trade-off between respondents' concern for avoiding misinformation outlets and their desire to improve advertising performance, respectively. Participants are told that they will be provided with the information they choose at the end of this study. Following the literature in measuring information acquisition[97], we measure respondents' demand for solution information, which serves as a revealed-preference proxy for their interest in implementing a solution for their organization.

Additionally, to measure whether the information treatment increases concern for financing misinformation in general, we record a second behavioural measure. Participants are told that the research team will donate US$100 to one of two organizations after randomly selecting one of the first hundred responses: (1) the GDI; and (2) DataKind, which helps mission-driven organizations increase their impact by unlocking their data science potential ethically and responsibly.

**Tackling experimental validity concerns.** Similarly to our consumer experiment, this survey was carried out in an online setting, where experimenter demand effects are limited[80,81]. We followed best practices[9] by keeping the treatment language neutral and ensuring the anonymity of the participants wherever possible. We find that most participants believe that the information provided in the survey was unbiased. Only about 7% of participants chose one of the 'biased' or 'very biased' options when asked to rate the political bias of the survey information provided from a seven-point scale ranging from 'very right-wing biased' to 'very left-wing biased'.

Importantly, to ensure truthful reporting, our main experimental outcomes were incentive-compatible. In particular, respondents who chose our platform solution demand outcome to learn about which platforms least contribute to placing companies' advertisements on misinformation websites had to face a trade-off between receiving this information and receiving information on improving advertising performance. Additionally, our baseline information demand outcomes elicited before the information intervention were also incentive-compatible in that participants would be asked to follow up on their decisions whether they opted for additional information via email or via an online information session.

These design choices are made to minimize demand effects on our main outcomes of interest. However, it is possible that these effects are still relevant, partially because participants may have an interest in 'doing the right thing' on a survey administered by an institution they have a connection with. We measure the relevance of potential demand effects using a survey question mirroring the approach used for our consumer experiment. To measure potential differences in the respondents' perceptions of the study across our treatment and control groups, we predict treatment status based on respondents' open-ended text responses about the purpose of the study via a support vector machine classifier, keeping 75% of the sample for the training set and the remaining 25% as the test set. We find that the classifier is only slightly worse than random chance in predicting treatment status (Supplementary Table 16) but similar in magnitude to those in the consumer experiment. Therefore, although experimenter demand effects may still be present, these results suggest that these effects do not drive our findings.

We address the external validity of our findings by verifying the decision-making capacity of our respondents within their organizations and by examining the generalizability of our sample. We find that the vast majority of those whose job titles we verify (94%) serve in executive or managerial roles within their organizations. The regression estimates in Supplementary Tables 18 and 19 show that our results remain qualitatively and quantitatively similar after the exclusion of the small sample of individuals in non-executive and non-managerial roles. Moreover, the verified and self-reported decision-makers are similar across observable characteristics as reported in Supplementary Table 17, suggesting limited selection in our verification process. To examine the generalizability of our sample, we investigate their observable characteristics. As shown in Supplementary Fig. 5, our sample of participants represents a broad array of company sizes and experience levels at their current roles. Additionally, about 22% of the executives in our sample (and 25% of all our participants) are women, which is aligned with the 21% to 26% industry estimates of women in senior roles globally[95,96].

### Reporting summary

Further information on research design is available in the Nature Portfolio Reporting Summary linked to this article.

## Data availability

Our study was pre-registered at the American Economic Association's Registry under AEARCTR-0009973. The data that we collected for our experimental studies are available in anonymized form and can be accessed from https://github.com/wajeeha-ahmad/misinformation-advertising. Data on job titles for the second survey experiment are not available, to protect participant confidentiality. Data analysing the descriptive analysis of advertising on misinformation websites can be made available after obtaining permission from the proprietary sources on misinformation domains (NewsGuard and the GDI) and advertising (Oracle). Source data are provided with this paper.

## Code availability

Code supporting the findings of the paper is available at https://github.com/wajeeha-ahmad/misinformation-advertising. Code analysing the descriptive analysis of advertising on misinformation websites can be made available after obtaining permission from the proprietary sources.

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

**Acknowledgements** The authors thank their data partners (NewsGuard, the GDI and Oracle) for sharing data; the executive education programmes at the Stanford Graduate School of Business and Heinz College at Carnegie Mellon University for partnership; T. Le for providing research assistance on part of this work; G. Jin, J. Wu, M. Liu, M. Collis, M. Gentzkow, P. Schwardmann, R. J. Duran, R. Moore, R. Appel, S. Agarwal, S. Borwankar and W. Lee for feedback and comments on earlier versions of this work; and participants at the Queen's Workshop on the Economics of Media 2022, 2022 MIT Conference on Digital Experimentation, Workshop on Information Systems Economics 2022, Rotman School of Management PhD Strategy Seminar 2023, ISMS Marketing Science Conference 2023, The Sumantra Ghoshal Conference at London Business School 2023, Platform Strategy Research Symposium at Boston University 2023, NBER SI 2023 Digital Economics and Artificial Intelligence, Social Science Research Council Workshop on the Economics of Social Media 2023, West Coast Research Symposium 2023, Stanford University Trust and Safety Research Conference 2023, Conference on Information Systems and Technology 2023, HKU Business School Management and Strategy Seminar 2023, and Boston University Online Research Seminar on Digital Businesses 2024. This research was supported in part by the Stanford Digital Economy Lab, Stanford McCoy Family Center for Ethics in Society, Stanford Impact Labs, Stanford Technology Ventures Program, Project Liberty Institute and the Economics of Digital Services initiative at the University of Pennsylvania.

**Author contributions** W.A. conceived the research and collected the descriptive data. W.A. and A.S. designed the survey experiments. W.A. conducted the consumer and decision-maker experiments. W.A. analysed the descriptive and experimental data for all three studies, with A.S. aiding in research conceptualization and data analysis. W.A. wrote the paper with input from A.S. W.A., A.S. and C.E. edited the paper. A.S., E.B. and C.E. supervised the work. All authors secured grant financing and partnerships, made revisions and approved the final manuscript.

**Competing interests** W.A. was a research intern at Microsoft during summer 2023. The other authors declare no competing interests.

**Additional information**
**Correspondence and requests for materials** should be addressed to Wajeeha Ahmad.

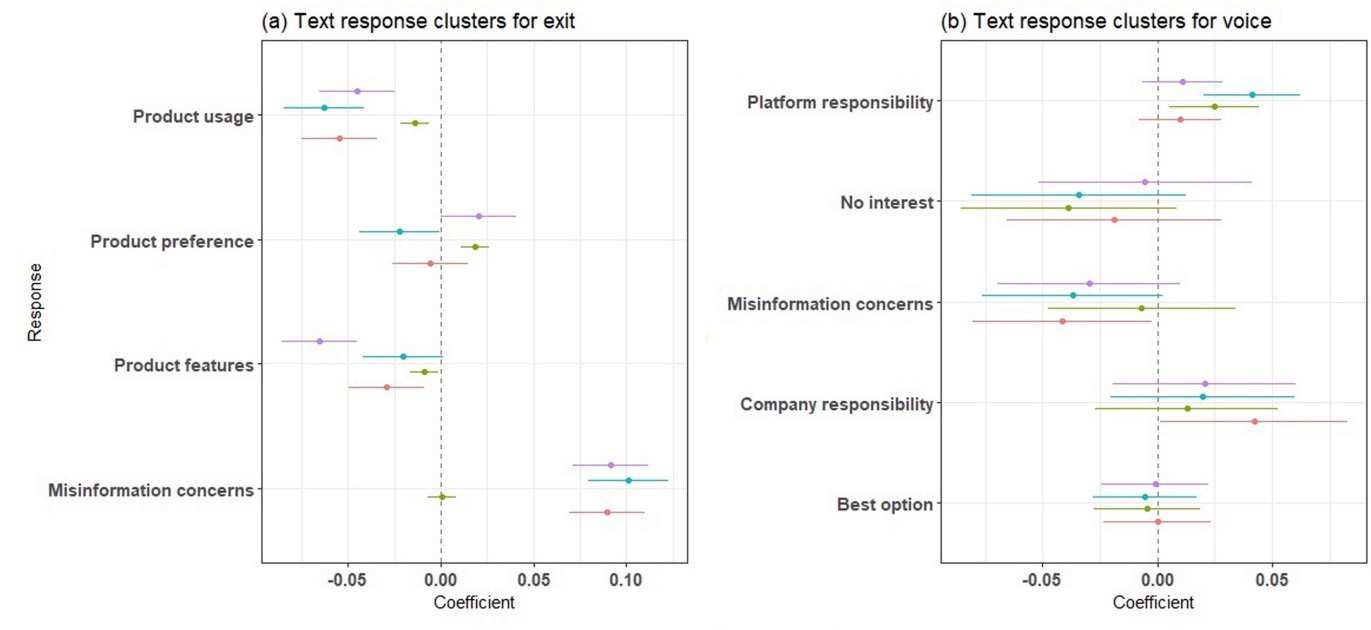

**Extended Data Fig. 1 | Text Explanation Clustering by Randomized Treatment Group.** *Notes:* This figure plots regression coefficients from OLS regressions of an indicator for cluster membership on each randomized group. Results are shown for our primary model specification, which controls for participants' demographic and behavioral characteristics (see Supplementary Information, "Analysis: Consumer study outcomes"). Data are presented as coefficients with the horizontal bars representing 95% confidence intervals derived from robust standard errors. The topics along the y-axes are binary variables that take value 1 if a participant's response is classified into the given topic and zero otherwise. Details about the text analyses are mentioned in Supplementary Information and sample text responses are shown in Tables A1 and A2. Figure (a) shows OLS regression results for text analysis on the open-ended reasons participants mentioned while explaining their choice of gift card ($n = 4039$). Figure (b) shows OLS regression results for text analysis on the open-ended reasons participants mentioned while explaining their choice of online petition to sign ($n = 4039$).

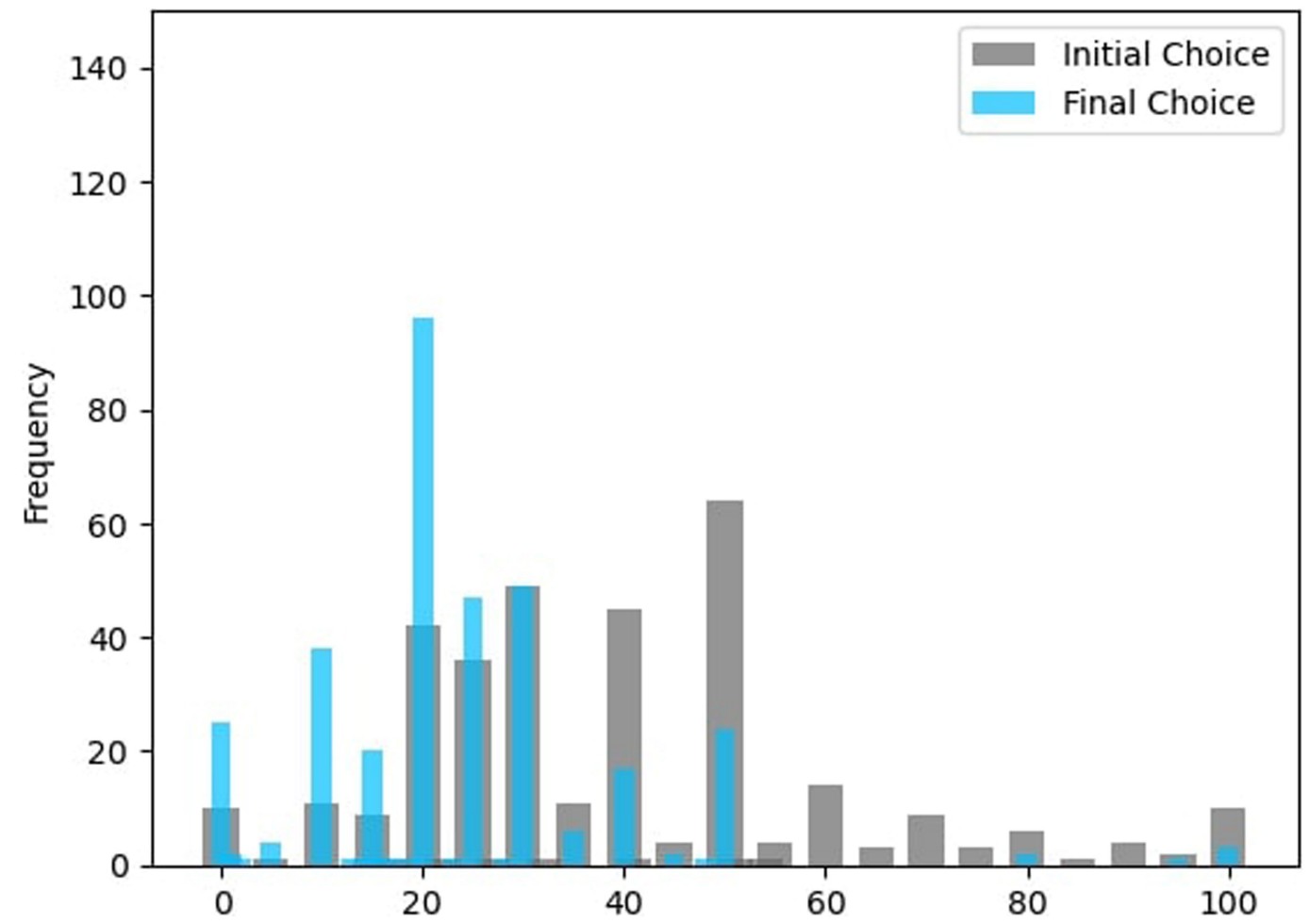

**Extended Data Fig. 2 | Weights Assigned by Treated Participants to their Initial and Final Gift Card Choices.** *Notes:* This figure shows the distribution of weights assigned by treated participants (i.e. those in randomized treatments T1, T3 or T4) to their top choice gift card before and after receiving the information treatment. The mean weight drops from 39.11 to 23.71 after receiving the information treatment, representing a 39.4% decline in the mean gift card value. The median weight drops from 35.0 to 20.0 after receiving the information treatment, representing a 42.9% decline in the median gift card value.

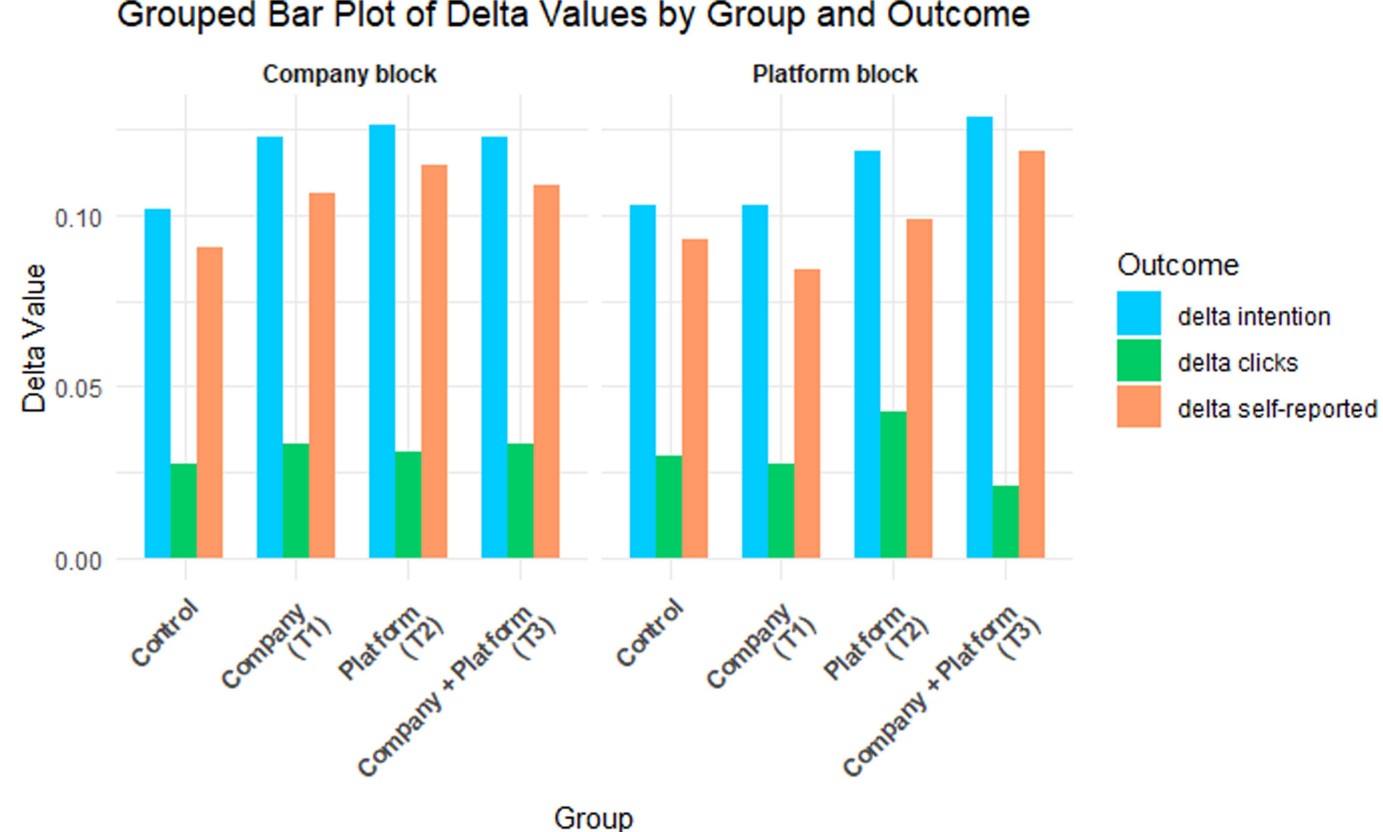

**Extended Data Fig. 3 | Delta Values by Treatment Group and Type of Voice Outcome.** *Notes:* The delta value reported here represents the difference between the proportion of a particular outcome variable and the proportion of actual recorded signatures for each treatment group.

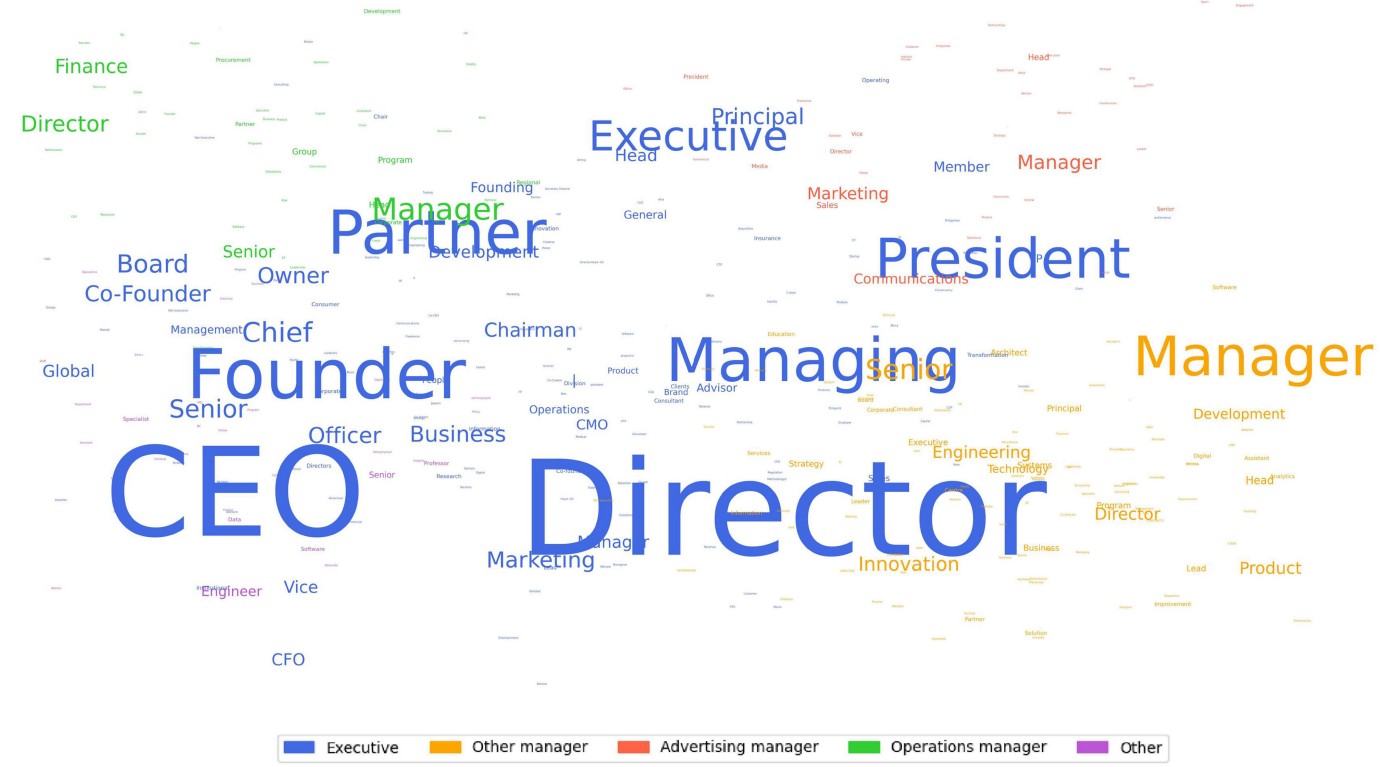

**Extended Data Fig. 4 | Verified job titles of participants in our second survey experiment by category.** *Notes:* This figure shows the job titles of the sub-sample of participants (N = 316) whose job titles we were able to verify from external sources, e.g., LinkedIn, Crunchbase, etc. The size of each word corresponds to its frequency of appearance in our sample.

**Extended Data Table 1 | Top 100 advertisers' activity on misinformation websites**

| | Appears on misinformation websites | | | Number of misinformation websites | | |
|---|---|---|---|---|---|---|
| | (1) | (2) | (3) | (4) | (5) | (6) |
| Digital ad platform usage | 3.36*** | 3.07*** | 3.41*** | 3.14*** | 2.82*** | 2.83*** |
| | (0.06) | (0.06) | (0.08) | (0.07) | (0.07) | (0.08) |
| | < 0.001 | < 0.001 | < 0.001 | < 0.001 | < 0.001 | < 0.001 |
| | | | | | | |
| Odds Ratio / Incidence Rate Ratio | 28.71 | 21.63 | 30.35 | 23.19 | 16.76 | 17.03 |
| Month FE | Yes | Yes | Yes | Yes | Yes | Yes |
| Industry FE | No | Yes | Yes | No | Yes | Yes |
| Month FE × Industry FE | No | No | Yes | No | No | Yes |
| Number of Observations | 11,969 | 11,969 | 11,969 | 11,969 | 11,969 | 11,969 |

*** $p < 0.01$; ** $p < 0.05$; * $p < 0.1$

This table shows regression results for the top 100 most active advertisers in a given year based on weekly data collected for the years 2019 to 2021 (*n*=11,969). In Columns (1)-(3), we show results from logit regressions where the dependent variable is *appears on misinformation websites,* a binary variable that takes value 1 when an advertiser appears on a misinformation website in a given week and zero otherwise. In Columns (4)-(6), we show results from poisson regressions where the dependent variable is *number of misinformation websites,* a continuous variable for the number of misinformation websites an advertiser appears on in a given week. *Digital platform usage* is a binary variable that takes value 1 when a company is using digital advertising platforms in a given week and zero otherwise. No adjustments were made for multiple comparisons. Robust standard errors in parentheses. P-values derived from two-sided t-tests reported below standard errors.

**Extended Data Table 2 | Comparison of responses across all petition outcomes**

| | Company | | | | Platform | | | |
|---|---|---|---|---|---|---|---|---|
| | Intention (1) | Clicks (2) | Reported (3) | Signed (4) | Intention (5) | Clicks (6) | Reported (7) | Signed (8) |
| Company (T1) | 0.03* | 0.02 | 0.03 | 0.02 | −0.01 | −0.02 | −0.02 | −0.02 |
| | (0.02) | (0.02) | (0.02) | (0.02) | (0.02) | (0.02) | (0.02) | (0.01) |
| | 0.098 | 0.257 | 0.132 | 0.272 | 0.453 | 0.349 | 0.205 | 0.307 |
| | | | | | | | | |
| Platform (T2) | 0.00 | −0.01 | 0.00 | −0.01 | 0.05** | 0.05*** | 0.04** | 0.04** |
| | (0.02) | (0.02) | (0.02) | (0.02) | (0.02) | (0.02) | (0.02) | (0.02) |
| | 0.937 | 0.436 | 0.908 | 0.361 | 0.015 | 0.006 | 0.048 | 0.026 |
| | | | | | | | | |
| Company and Platform (T3) | 0.01 | −0.00 | 0.01 | −0.01 | 0.03 | −0.01 | 0.03 | 0.00 |
| | (0.02) | (0.02) | (0.02) | (0.02) | (0.02) | (0.02) | (0.02) | (0.02) |
| | 0.581 | 0.940 | 0.600 | 0.566 | 0.194 | 0.708 | 0.196 | 0.956 |
| | | | | | | | | |
| Company Ranking (T4) | 0.04* | 0.04** | 0.03 | | −0.03 | −0.03 | −0.03 | |
| | (0.02) | (0.02) | (0.02) | | (0.02) | (0.02) | (0.02) | |
| | 0.055 | 0.048 | 0.145 | | 0.126 | 0.115 | 0.106 | |
| | | | | | | | | |
| Controls | Yes | Yes | Yes | No | Yes | Yes | Yes | No |
| Control mean | 0.22 | 0.15 | 0.21 | 0.12 | 0.21 | 0.14 | 0.20 | 0.11 |
| Observations | 4039 | 4039 | 4039 | 3225 | 4039 | 4039 | 4039 | 3225 |

$^{***}p < 0.01$, $^{**}p < 0.05$, $^{*}p < 0.1$

*Notes:* This table shows OLS regression results for each of the four treatment groups (T1, T2, T3 and T4) across all of our petition outcomes. Columns (1) to (4) refer to company-specific petitions suggesting that companies like the respondent's top choice gift card company need to block their advertisements from appearing on misinformation websites. Columns (5) to (8) refer to platform-specific petitions suggesting that digital advertising platforms used by companies need to block advertisements from appearing on misinformation websites. In columns (1) and (5), the dependent variable is the intention to sign a petition, a binary variable that takes the value 1 when a participant indicates wanting to sign a given petition and zero otherwise ($n$=4039). In columns (2) and (6), the dependent variable is a click on the petition link that takes the user to the Change.org platform to sign a petition, a binary variable that takes the value 1 when a participant clicks on the link and zero otherwise ($n$=4039). In columns (3) and (7), the dependent variable is the self-reported petition signature, a binary variable that takes the value 1 when a participant reports having signed a given petition and zero otherwise ($n$=4039). We record actual petition signatures in columns (4) and (8); we omit signatures for the T4 group since these petitions were accidentally deleted by Change.org ($n$=3225). Since we only observe actual signatures on the treatment group level, we cannot include controls and run regressions for these outcomes. To do testing, we calculate standard errors using the standard formula for (two-sided) proportion tests. For the remaining columns, we apply robust standard errors in parentheses. No adjustments were made for multiple comparisons. P-values derived from two-sided t-tests reported below standard errors.

**Extended Data Table 3 | Heterogeneous Treatment Effects for Exit and Voice**

| | Switch in gift card from top choice company ("exit") | | | | Petition clicks on company petition ("voice") | | | |
|---|---|---|---|---|---|---|---|---|
| | (1) | (2) | (3) | (4) | (5) | (6) | (7) | (8) |
| Treatment | 0.07*** | 0.07*** | 0.12*** | 0.10*** | 0.02 | 0.00 | 0.03** | 0.03** |
| | (0.01) | (0.01) | (0.01) | (0.01) | (0.02) | (0.02) | (0.02) | (0.01) |
| | < 0.001 | < 0.001 | < 0.001 | < 0.001 | 0.127 | 0.887 | 0.040 | 0.012 |
| Treatment × Female | 0.05** | | | | | 0.00 | | |
| | (0.02) | | | | | (0.02) | | |
| | 0.011 | | | | | 0.980 | | |
| Treatment × Biden voter | | 0.03* | | | | 0.05** | | |
| | | (0.02) | | | | (0.02) | | |
| | | 0.058 | | | | 0.036 | | |
| Treatment × Frequent user | | | −0.05*** | | | | −0.02 | |
| | | | (0.02) | | | | (0.02) | |
| | | | 0.007 | | | | 0.486 | |
| Treatment × Consumes misinformation | | | | −0.04* | | | | −0.03 |
| | | | | (0.02) | | | | (0.03) |
| | | | | 0.097 | | | | 0.231 |
| Female | 0.00 | 0.03*** | 0.03*** | 0.03*** | 0.00 | 0.01 | 0.01 | 0.00 |
| | (0.01) | (0.01) | (0.01) | (0.01) | (0.02) | (0.01) | (0.01) | (0.01) |
| | 0.681 | 0.002 | 0.002 | 0.002 | 0.785 | 0.674 | 0.667 | 0.686 |
| Biden voter | 0.01 | −0.01 | 0.01 | 0.01 | 0.02* | −0.01 | 0.02* | 0.02* |
| | (0.01) | (0.01) | (0.01) | (0.01) | (0.01) | (0.02) | (0.01) | (0.01) |
| | 0.579 | 0.257 | 0.576 | 0.587 | 0.097 | 0.708 | 0.095 | 0.096 |
| Frequent user | −0.04*** | −0.04*** | −0.01 | −0.04*** | 0.03** | 0.03** | 0.04** | 0.03** |
| | (0.01) | (0.01) | (0.01) | (0.01) | (0.01) | (0.01) | (0.02) | (0.01) |
| | 0.001 | 0.001 | 0.611 | 0.001 | 0.014 | 0.014 | 0.026 | 0.014 |
| Consumes misinformation | 0.03** | 0.03** | 0.03** | 0.05*** | 0.00 | 0.00 | 0.00 | 0.02 |
| | (0.01) | (0.01) | (0.01) | (0.02) | (0.01) | (0.01) | (0.01) | (0.02) |
| | 0.011 | 0.013 | 0.014 | 0.001 | 0.893 | 0.894 | 0.902 | 0.316 |
| Controls | Yes | Yes | Yes | Yes | Yes | Yes | Yes | Yes |
| Control group mean | 0.05 | 0.05 | 0.05 | 0.05 | 0.14 | 0.14 | 0.14 | 0.14 |
| Observations | 4039 | 4039 | 4039 | 4039 | 4039 | 4039 | 4039 | 4039 |

***$p < 0.01$, **$p < 0.05$, *$p < 0.1$

Notes: This table shows OLS regression results for our full sample ($n=4039$) where *Treatment* is a binary variable that takes a value of 1 if a respondent is randomized into any of the company-specific treatment groups (T1, T3 or T4) and zero otherwise. In columns 1 to 4, the dependent variable is switch in gift card choice from the respondent's top choice company (i.e. "exit"). In columns 5 to 8, the dependent variable is clicking on a link to sign a petition that suggests that companies like the respondent's top choice gift card company need to block their advertisements from appearing on misinformation websites. *Female* is a binary variable that takes a value of 1 if a respondent reports being female and zero otherwise. *Biden voter* is a binary variable that takes a value of 1 if a respondent reported voting for President Biden in the 2020 US Presidential election and zero otherwise. *Frequent user* is a binary variable that takes a value of 1 if a respondent reported using their top choice gift card at least once a month. *Consumes misinformation* is a binary variable that takes a value of 1 if a respondent reported using one or more misinformation news outlets (out of a list of 26 popular news outlets) in the past 12 months and zero otherwise. No adjustments were made for multiple comparisons. Robust standard errors in parentheses. P-values derived from two-sided t-tests reported below standard errors.

**Extended Data Table 4 | Decision-makers' Beliefs and Characteristics about Advertising on Misinformation Websites**

| | All (1) | Executives (2) | Managers (3) | Other (4) |
|---|---|---|---|---|
| Belief about Advertising on Misinformation | 0.20 | 0.18 | 0.24 | 0.17 |
| Certainty of Belief about Advertising on Misinformation | 0.79 | 0.81 | 0.75 | 0.83 |
| Advertised on Misinformation* | 0.81 | 0.72 | 0.86 | 1.00 |
| % of Correct Beliefs about Own Company * | 0.36 | 0.41 | 0.33 | 0.25 |
| Observations | 442 | 248 | 147 | 47 |

*Notes:* This table shows respondents' beliefs and characteristics about their own company advertising on misinformation outlets. Column (1) shows results for the full sample (*n*=442), Column (2) for the sub-sample of executives (*n*=248), Column 3 for the sub-sample of managers (*n*=147), and Column 4 for the remaining individuals (*n*=47). The proportions in rows marked with an asterisk (*) are calculated based on the subsample of participants who requested an ad check and whose companies appeared in our advertising data (*n*=106) in Column (1).

# Reporting Summary

## Statistics

For all statistical analyses, confirm that the following items are present in the figure legend, table legend, main text, or Methods section.

| n/a | Confirmed | |
|---|---|---|
| ☐ | ☒ | The exact sample size (*n*) for each experimental group/condition, given as a discrete number and unit of measurement |
| ☐ | ☒ | A statement on whether measurements were taken from distinct samples or whether the same sample was measured repeatedly |
| ☐ | ☒ | The statistical test(s) used AND whether they are one- or two-sided *Only common tests should be described solely by name; describe more complex techniques in the Methods section.* |
| ☐ | ☒ | A description of all covariates tested |
| ☐ | ☒ | A description of any assumptions or corrections, such as tests of normality and adjustment for multiple comparisons |
| ☐ | ☒ | A full description of the statistical parameters including central tendency (e.g. means) or other basic estimates (e.g. regression coefficient) AND variation (e.g. standard deviation) or associated estimates of uncertainty (e.g. confidence intervals) |
| ☐ | ☒ | For null hypothesis testing, the test statistic (e.g. *F*, *t*, *r*) with confidence intervals, effect sizes, degrees of freedom and *P* value noted *Give P values as exact values whenever suitable.* |
| ☒ | ☐ | For Bayesian analysis, information on the choice of priors and Markov chain Monte Carlo settings |
| ☐ | ☒ | For hierarchical and complex designs, identification of the appropriate level for tests and full reporting of outcomes |
| ☐ | ☒ | Estimates of effect sizes (e.g. Cohen's *d*, Pearson's *r*), indicating how they were calculated |

*Our web collection on statistics for biologists contains articles on many of the points above.*

## Software and code

Policy information about availability of computer code

| | |
|---|---|
| Data collection | Data on advertising was obtained using Oracle's Moat Pro platform, which collects data by scraping thousands of websites each day. Data on companies and ad platforms was manually extracted from this platform from January 1, 2019 to December 31, 2021. Data collection for both survey experiments was done using the university provided Qualtrics survey software. Code supporting the findings of the paper is available at: https://github.com/wajeeha-ahmad/misinformation-advertising. |
| Data analysis | Data was cleaned and partially analyzed in Python using Jupyter notebooks. Analysis was completed using R version 4.0.3. |

For manuscripts utilizing custom algorithms or software that are central to the research but not yet described in published literature, software must be made available to editors and reviewers. We strongly encourage code deposition in a community repository (e.g. GitHub). See the Nature Portfolio guidelines for submitting code & software for further information.

## Data

Policy information about availability of data

All manuscripts must include a data availability statement. This statement should provide the following information, where applicable:

- Accession codes, unique identifiers, or web links for publicly available datasets
- A description of any restrictions on data availability
- For clinical datasets or third party data, please ensure that the statement adheres to our policy

Our study was preregistered at the American Economic Association's Registry under AEARCTR-0009973. The data we collected for our experimental studies is

available in anonymized form and can be accessed by clicking on this link: https://github.com/wajeeha-ahmad/misinformation-advertising. Data on job titles for the second survey experiment are not available to protect participant confidentiality. Data analyzing the descriptive analysis of advertising on misinformation websites can be made available after obtaining permission from the proprietary sources on misinformation domains (NewsGuard and the Global Disinformation Index) and advertising (Oracle).

# Research involving human participants, their data, or biological material

Policy information about studies with underline{human participants or human data}. See also policy information about underline{sex, gender (identity/presentation), and sexual orientation} and underline{race, ethnicity and racism}.

| | |
|---|---|
| Reporting on sex and gender | Participants in both surveys were asked to self-report their gender as part of the survey using a single question (Q. What is your gender? Response categories: Male, Female, Non-binary or third gender, Prefer not to say). Gender representation was considered for the consumer survey to ensure a nationally representative sample of the U.S. population based on gender among other dimensions. While our findings apply to all participants regardless of gender, we report pre-registered gender-specific results for our consumer experiment in the section titled "Heterogeneous treatment effects". Gender-based analysis was not performed for the decision-maker study given its small sample size. Our consumer sample consisted of 52% female participants and our decision-maker sample consisted of 21% female participants. |
| Reporting on race, ethnicity, or other socially relevant groupings | Participants in our consumer survey were asked to self-report their race or ethnicity using a single survey question (Q: which of the following best describes your ethnicity or race? Response categories: Asian/Asian American, Caucasian/White, Native American/Inuit/Aleut, Native Hawaiian/Pacific Islander, Other, Prefer not to say.) Race was considered in the survey design to ensure a nationally representative sample of the U.S. population in terms of race among other dimensions. We also controlled for the respondents' race category chosen in response to the above question while performing our analyses. No data on race was collected for the decision-maker study. |
| Population characteristics | See the "Behavioral and social sciences study design" responses below. Population characteristics are further detailed in the Methods section of our paper. Summary statistics for our participant populations are also reported in Supplementary Information Tables A5 and A11. |
| Recruitment | Participants in our consumer survey were recruited via CloudResearch. These participants were invited to "take a survey about the news, technology and businesses." This generic description does not make any specifc references to misinformation or its effects to avoid self-selection based on interest in or perceptions of misinformation. Participants may have self-selected into taking our survey based on their broad interest in news, technology and/or businesses; given the expected prevalence of such general interests among the consumers of the advertising companies we study, we do not expect such self-selection to substantially bias our results.

Participants in our decision-maker survey were recruited via emails sent by our partner organizations. We used neutral language in our study recruitment emails to attract a broad audience of participants to our survey regardless of their initial beliefs and concerns about misinformation, stating our goal as "conducting vital research on the role of digital technologies in impacting your organization" without mentioning misinformation. While our sample is limited to those decision-makers who participated in executive education programs at our two partner programs, we find the industries these decision-makers were from to be representative of the industries we observed in our descriptive analysis of advertisers appearing on various misinformation websites. Further details about the representativeness and validity of our sample are provided in Methods, "Decision-maker experiment design: Tackling experimental validity concerns". |
| Ethics oversight | The study was reviewed by the Stanford University Institutional Review Board (Protocol No. IRB-63897) and the Carnegie Mellon University Institutional Review Board (Protocol No. IRB00000603). |

Note that full information on the approval of the study protocol must also be provided in the manuscript.

# Field-specific reporting

Please select the one below that is the best fit for your research. If you are not sure, read the appropriate sections before making your selection.

☐ Life sciences ☒ Behavioural & social sciences ☐ Ecological, evolutionary & environmental sciences

For a reference copy of the document with all sections, see nature.com/documents/nr-reporting-summary-flat.pdf

# Behavioural & social sciences study design

All studies must disclose on these points even when the disclosure is negative.

| | |
|---|---|
| Study description | The study is quantitative, involving descriptive analyses and experimental survey data. Study designs are outlined in the Methods section of the paper. |
| Research sample | The sample for the consumer study was provided by CloudResearch, which recruited participants in the U.S. based on quotas to ensure a nationally representative sample for the U.S. population on three criteria: gender, age and race. CloudResearch was chosen for its higher quality participant pool (relative to MTurk) based on prior analysis and its ability to provide a large sample of consumers of commonly used products/services in the U.S. Further details about the consumer study research sample are provided in Methods, "Consumer experiment design: Setting and sample recruitment" and in Supplementary Information, "Section 2.2: Consumer study results". |

| | |
|---|---|
| | The sample for the decision-maker study is from the alumni pool of Executive Education programs at two of our partner organizations. As mentioned in the Methods section, this sample was chosen in order to survey senior managers and leaders who could influence strategic decision-making within their firms. Additionally, partnering with two university programs instead of a specific firm allowed us to access a more diverse sample of companies than prior work that sampled specific types of firms, e.g. innovative firms, startups or small businesses. urther details about the decision-maker study research sample are provided in Methods, "Decision-maker experiment design: Setting and sample recruitment" and in Supplementary Information, "Section 2.3: Decision-maker study results". |
| Sampling strategy | CloudResearch collected data for our consumer study, ensuring that the sample was nationally representative for the U.S. population based on three criteria: gender, age and race. We performed power calculations to arrive at a rough estimate of the sample size required.<br><br>Data for our decision-making study was collected by sending invite emails with the survey link via our partner organizations, who randomly sampled potential study participants from their alumni pool. The sample size we arrived at was based on the maximum sample our partner organizations were comfortable sending emails to. |
| Data collection | Survey data was collected using Qualtrics survey software. Respondents could complete the survey on their own (without any researchers present at the time of them filling out the survey) using any appropriate web- and browser-enabled device. |
| Timing | Data for our consumer study was collected between August 29, 2022 and September 7, 2022.  An initial data sample for our decision-maker study was collected from November 22, 2022 to December 5, 2022 with a larger sample being collected from December 7, 2022 to December 26, 2022. |
| Data exclusions | In both experiments, participants were excluded from continuing the survey if they did not provide consent to participate in the study at the beginning of the survey. These included 483 participants in our consumer survey and 8 participants in our decision-maker survey.<br><br>Participants in our consumer survey were further restricted from continuing the study if they reported not being a U.S. citizen or being based in the U.S. This excluded 188 participants.<br><br>59 participants in the decision-maker study were further excluded if they reported not being employed at the time of the survey.<br><br>Finally, participants in both experiments were excluded from analyses if they exhibited inattentiveness during our survey by incorrectly answering specific question(s). This excluded 5609 participants from our consumer study and 66 from our decision-maker study.<br><br>These exclusion criteria were pre-established. |
| Non-participation | In our consumer experiment, 483 participants declined participation by not providing consent to participate at the beginning of the survey. A further 188 participants were dropped out from the study after they reporting not being a U.S. citizen or not being based in the U.S. Finally, 5609 participants were dropped from analyses for incorrectly answering our attention check question.<br><br>In our decision-maker study, 8 participants declined participation by not providing consent at the beginning of the survey. 59 participants in the decision-maker study were dropped out if they reported not being employed at the time of the survey. Finally, 66 participants were dropped from analyses for incorrectly answering our attention check questions. |
| Randomization | Randomization for both survey experiments was completed using Qualtrics survey software. |

# Reporting for specific materials, systems and methods

We require information from authors about some types of materials, experimental systems and methods used in many studies. Here, indicate whether each material, system or method listed is relevant to your study. If you are not sure if a list item applies to your research, read the appropriate section before selecting a response.

## Materials & experimental systems

| n/a | Involved in the study |
|---|---|
| ☒ | ☐ Antibodies |
| ☒ | ☐ Eukaryotic cell lines |
| ☒ | ☐ Palaeontology and archaeology |
| ☒ | ☐ Animals and other organisms |
| ☒ | ☐ Clinical data |
| ☒ | ☐ Dual use research of concern |
| ☒ | ☐ Plants |

## Methods

| n/a | Involved in the study |
|---|---|
| ☒ | ☐ ChIP-seq |
| ☒ | ☐ Flow cytometry |
| ☒ | ☐ MRI-based neuroimaging |

## Plants

Seed stocks

N/A

Novel plant genotypes

N/A

Authentication

N/A

