## [Peer Review File · Nature]

Manuscript Title: Companies inadvertently fund online misinformation despite consumer backlash

Reviewer Comments & Author Rebuttals

Reviewer Reports on the Initial Version:

Referee #1 (Remarks to the Author):

Referee report on “Combating Misinformation: A Supply-side Approach”

Short summary of the paper

The authors conduct impressive data collection efforts in three parts. First, they collect descriptive evidence on the extent of advertising on misinformation pages by major companies. Second, they conduct an information provision experiment with a general population sample to see whether consumers punish companies who advertise on misinformation pages. Third, they conduct another information provision experiment with a premium sample of firm decision-makers to see how firms respond to information about misinformation advertising and their demand for information relevant to advertising and misinformation pages. Their main results show 1) that advertising on misinformation pages is surprisingly common, partly driven by digital ad platforms, 2) that consumers punish firms for advertising on misinformation pages, irrespective of whether or not firms use digital ad platforms, and 3) that firm decision-makers are often unaware of their own advertising on misinformation pages; underestimate the role played by digital ad platforms; and have a high demand for information relevant for reducing their ad exposure to misinformation sites.

Evaluation of the main contribution

I think this is a great paper on an important topic. I was enthusiastically reading the paper and felt I learned a lot from reading it. I also want to applaud the authors for very carefully collecting rich descriptive data on the extent of firm advertising on misinformation pages as well as conducting two information provision experiments that cover both the consumer and firm sides. The paper is also on a very important topic and, while there is not much “conceptual” novelty in terms of methods or theory development, I very much agree with the authors that the supply-side approach they push has been underexplored in the literature. While I have a lot of enthusiasm for the paper, I also have a few comments and potential criticism that I outline below.

Main comments

#1. An important question is to what extent the gift card experiment is an externally valid measure of consumer responses to misinformation ads. The respondents can initially indicate their preference between six gift cards from most preferred to least preferred. Post-treatment, they can then select which gift card they actually want to receive. The main outcome is whether people choose a different gift card than their top choice (i.e., “switch”, as reported in columns 1 and 2 of Table 1). Depending on how much they value their first choice relative to the other choices, the decision to “switch” can be quite costly or not costly at all. The authors measure the weights of each gift card (which must sum to 100), so they have data on the intensity of the preference, and hence the respondents’ relative valuation of each gift card. I think it’s important to investigate how costly it was for people to switch from their top choice. For instance, they could calculate how many percent of the gift card value they lose by switching from their most preferred option to a different option (using the relative weights). E.g. if Uber gets a weight of 90 and Lyft gets a weight of 10, switching from Uber to Lyft reduces the implied gift card value much more than if Uber gets 55 and Lyft gets 45. Furthermore, the authors could acknowledge somewhere that they measure short-term effects and it is not clear how large the effect would be e.g. one month later when the issue

was less salient. That is, it could be the case that consumers immediately punish a company for placing ads on a misinformation page, but after some time, they forget about it and it is “business as usual”. A follow-up study to test whether the effect persists would have been very interesting (in which I think Likert scale outcomes would have been totally fine).

#2. I think the authors are a bit too dismissive of the value of self-reported measures in section 3.2.4, mostly because I think the Likert scale outcome measures different things. A more direct comparison between the behavioral outcomes and self-reported/Likert scale outcomes would be along the lines of “How likely are you to use Uber in the future?”.

#3. The authors write that the “platform only” treatment group (T2) “practically serves as a second control group for our company-level outcomes”, yet, relative to the control group, they see a 3pp increase in their main outcome (columns 1 and 2 of Table 1) for T2 relative to the control group. I think it’s worth commenting/discussing this result in the main text.

#4. The authors use petition links as their primary “voice” outcome, which I think is a nice and relevant outcome measure. However, I think it’s somewhat problematic that they focus on link clicks - which, I believe, could be quite noisy. I think the intention to sign, reported signatures and actual signatures are all preferable to link clicks (or the link clicks should be validated somehow). The authors also write that their results are robust to using petition intentions or actual signatures, but they don’t have the petition signatures for the T4 treatment (the table footnote for the appendix table reveals that these petitions were accidentally deleted). I think this is worth highlighting in the main text. Another thing that is interesting about Table A6 is that self-reported signatures and actual signatures are almost identical (control group mean of 15% and 14%, respectively) - this actually speaks to the value of self-reported data. In the discussion of Table 2, I also think the authors should discuss the null findings as well and not only focus on the statistically significant effects.

#5. Table 6 needs more documentation. The authors focus on the subsample who reported “No” to think that their companies had ads on misinformation pages and then split this sample by whether they were “certain” or “uncertain” about this. Consulting the instructions, it seems that the question of certainty was elicited on a 5-point Likert scale from “Very sure” to “Very unsure”. It is unclear whether those who were “certain” include those who said “Very sure” or also those who said “Sure”. It also leaves me a little bit uncomfortable that, splitting the sample this way, they end up with only 68 observations in columns 2/4 of Table 6 to find a 40% p.p. treatment effect. Is this effect robust to different splits of the “Sure/Unsure” variable? Was the split they decided on pre-specified? To find statistically significant effects with only 68 observations, they need really large treatment effects, which is somewhat problematic.

Minor comments

#1. On p. 13, the authors write that the treatment effect of T4 amounts to “seven percentage points”, yet the number in columns 1-2 of Table 2 shows an 8 pp change. This discrepancy should be corrected (I assume the table version is correct).

#2. In section 4.1.1 “Setting and sample size”, the sample size is actually not included.

#3. In Table 1, I would include p-values for equality of treatment effects such as $T_2=T_3$, etc.

#4. In section 3.2.2, the authors report percent instead of percentage points change (36% more...) I think it makes sense to also include the pp change in the main text for better context.

#5. Footnote 23 on p. 20 says that they dropped respondents who answered greater than 100 when asked to estimate a number out of 100. I would clarify whether this was done pre-treatment or post-treatment (post-treatment is problematic because the treatment could theoretically induce people to become more/less attentive).

Referee #2 (Remarks to the Author):

This paper evaluates the possibility of a supply-side intervention to suppress the availability of misinformation (really: low quality websites) on the internet. It evaluates this with a 3 prong study. The first study evaluates the prevalence of advertising (and evaluation of which advertisers) on misinformation websites. The second evaluates whether consumers have a material preference for companies that do not advertise on misinformation websites. And the third evaluates whether decision makers at companies were aware of and would have a preference for avoiding advertising on such websites.

Overall, I quite like this paper. The issue of the financial undergirdings of misinformation sites is critical, but understudied. There is far more research on how to reduce demand for misinformation; yet it's not clear to me that many of those approaches scale. The potential for eliminating the business model for the production of misinformation is thus quite attractive as an approach to the broader issue.

The three independent analyses provide a compelling package. The first study uses a data set of ads scraped from a large set of misinformation sites, finding a wide array of advertisers present, and apparently largely driven by a small number of advertising platforms. These platforms separate buyers of ads from their placement on particular websites (though there are underutilized tools to exclude websites from particular campaigns). The second finds that for a representative sample, there are material preferences for avoiding products from companies that advertise on these websites. This is done through an information intervention, where people can choose among a set of gift cards to receive, and are then informed about whether the company associated with the gift card had advertised on misinfo sites (all had, but in varying amounts). The bottom line is that the intervention drove people away from companies that advertised on the misinfo sites, as well as to sign an online petition (there are a number of interesting nuances). The third takes a sample of business leaders (alums of Stanford and Carnegie Mellon exec ed), surveys them (and includes an information intervention there as well), finding strong preferences for ads not appearing on misinfo website (again, interesting nuances).

There are limitations, of course. The second and third studies have potential generalizability issues, in terms of whether this would truly hold in the field. But, I think they've taken steps to maximize the potential external validity, like make sure there were real (if not large) stakes for the subjects. I do have concerns in the third study that the alums might want to "do the right thing" for a study from the institutions that they had attended (such as attending is for exec ed, but still). For study 1, I'd like more explanation of the 10:1 figure re use of advertising platforms (a figure that is then used in the experimental studies, so it's especially important). For example, couldn't the 10:1 figure simply reflect that the companies that used the platforms just advertised more that week? I'm assuming not, but a little more explanation in interpreting these numbers would be helpful.

Other bits of housekeeping: it would be useful to know if there were any deviations from the PAPs (usually there would be; and this is especially important context for interpreting some of these heterogeneous treatment effects- were all of them preregistered?). Also, it is not clear why the analysis code would not be made available along with the data (as compared to upon request).

Referee #3 (Remarks to the Author):

This manuscript provides several pieces of novel, rigorous and useful evidence on the understudied "supply side" of online misinformation. Each one of them would be interesting on its own, and in combination they present a fairly coherent picture of platforms' role in serving ads on unreliable

websites, (potential) consumer backlash, and low awareness on behalf of corporate decision-makers. The descriptive evidence is comprehensive and clear, and the experiments are designed with a great deal of care to guard against cheap talk, demand effects, etc.

I have three general points about the paper. First, it seems to have been written primarily with an economics audience in mind (looking at citations etc.), which would be fine for an econ journal but not Nature. (Relatedly, the main text is far too long and detailed for a Nature-style article.) Second, the weakest link here is the "decision-maker" experiment, which -- while carefully and cleverly designed -- has questionable generalizability due to the peculiarities of the sample. Third, the policy recommendations that follow from this last experiment seem inadequate to address the problem. For example, the authors begin by summarizing the "significant social consequences" of misinformation, but the paper ends with a call for transparency and voluntary self-regulation, perhaps brought on by internal employee pressure. As with much of the literature in this space, there is a mismatch between the stated severity of the problem and the proposed solutions.

On the second experiment: I am skeptical that the sample comprises actual "decision-makers" at these companies. Do we know that people honestly self-reported being a CEO? Is there a way (e.g., LinkedIn) to bolster this claim? Otherwise another term should be used to describe the sample. I'm also concerned about possible inattentiveness driving some of the puzzling results (such as the null on donation preferences). I don't see that attention screeners were used for this experiment -- is that the case?

On the policy discussion: Are the respondents in the "decision-makers" sample actually empowered to make decisions, or are they simply another set of people (i.e., employees) who can exercise their voice within a company? That aside, I am somewhat puzzled that the paper simply stops here -- in other words, let's disseminate accurate information about the role of ad platforms to corporate executives, hope they do the right thing, and call it a day. As an example of what I mean, the Digital Services Act just came into force across the EU and could have enforcement mechanisms to make exactly these kinds of changes, so it's surprising to see no engagement with this or other relevant policy developments around the world.

Finally, a note on terminology. A lot of literature in this space uses terms like "unreliable" or "untrustworthy" to describe websites like those on the NewsGuard list. I'd strongly recommend adopting one of these throughout. There are a number of reasons for this, but most importantly, a lot of stories published by these websites are not strictly speaking false. Since the designations are at the publisher level, it makes sense to use a term that more closely describes the verification processes and overall commitment to accuracy.

Other points:

- 2.3.2: Approximately 8% of companies that do not use digital ad platforms appear on misinformation websites. Can you say more about these 8% of companies? Who are they, what distinguishes them?

- Possibly useful citation with related descriptive evidence: Bozarth, L., & Budak, C. (2021). An Analysis of the Partnership between Retailers and Low-credibility News Publishers. *Journal of Quantitative Description: Digital Media*, 1. <https://doi.org/10.51685/jqd.2021.010>

- Useful citation for demand effects: Mummolo, J. and Peterson, E., 2019. Demand effects in survey experiments: An empirical assessment. *American Political Science Review*, 113(2), pp.517-529.

Referee #4 (Remarks to the Author):

This paper explores the prevalence and impact of advertising on web pages that traffic in misinformation. Using observational data on website advertising patterns and two survey experiments – one on a sample of ordinary Americans and another on a convenience sample of advertising decision-makers from U.S. companies – the authors convincingly make the case that web advertising on misinformation websites is prevalent and can carry negative consequences for companies advertising on those webpages. At the same time, individuals involved in the decision-making process over advertising placement both underestimate the prevalence of placement of advertising on low-quality websites and are receptive to corrective information about the true level of advertising on low-quality websites. The authors therefore conclude that low-cost interventions that make explicit the prevalence and nature of advertising on low-quality websites placed through ad markets could increase the knowledge of such practices and encourage companies to reduce their financing of such advertising.

I think that this is a really great paper that should be published. Each of the constituent pieces (observational web data, survey experiment of ordinary Americans, survey experiment of decision makers) is well designed and well executed. The information treatments are well designed and the instrumentation used to gather belief and behavior data is well thought out. I especially like the behavioral measures employed by the authors (the gift-card preference measures in the ordinary Americans experiment and the petition signing in the decision maker survey, especially). These are clever and creative measures that allow the authors to get at the underlying mechanism of choice in a compelling way.

But even stronger than the constituent pieces is how the authors bring these pieces together to tell a compelling story about a matter of critical importance in current society – and provide actionable recommendations to address that problem. The observational web data sets the stage by measuring the amount and type of advertising on low-quality web pages (measured through domains identified by widely used and recognized independent sources, such as NewGuard, and leading academic research teams such as Guess et al and Allcott et al). The survey experiment of ordinary Americans estimates the larger impact of these advertising patterns on company reputation. Specifically, the experiment assesses the effect of the provision of information about both the advertising placement of companies' individuals support (indexed by a desire to get a gift card from a particular company in a raffle) and the impact of the use of digital ad platforms on overall patterns of advertising on low-quality websites. And the decision-maker survey demonstrates that companies place ads on low-quality websites, not because they are actively trying to do so, but instead because those decision-makers have an incorrect view of the actual state of affairs. By providing both consumers and advertisers with correct information about the nature of the advertising landscape – information gleaned through the observational studies – the authors convincingly demonstrate that the financing of low-quality websites is the result of unintended consequences driven by incomplete information. Once such information is provided, the supply of misinformation in the informational ecosystem could be reduced – and reduced substantially given the size of the effects estimated in the experiment.

I did have a couple minor critiques of the paper. First, the sample of ordinary Americans is not a "representative" sample. It is better thought of as a "diverse" sample (see, for example, the discussion of samples in this paper by Coppock and McClellan:

<https://journals.sagepub.com/doi/pdf/10.1177/2053168018822174>). I would change the language in the paper accordingly. Second, there were several points in the paper where the authors discussed discarding respondents who failed an attention check or gave numerically non-sensible answers. Discarding such respondents is not always the best strategy given patterns of attrition in surveys. For completeness would like to see a set of robustness checks in the appendix.

But these are very minor points. I very much enjoyed reading this paper and I learned a great deal. It is an exemplar of the kind of research in computational social science that can elucidate the problems of the day and I support publication.

Reviewer 1

Short summary of the paper

The authors conduct impressive data collection efforts in three parts. First, they collect descriptive evidence on the extent of advertising on misinformation pages by major companies. Second, they conduct an information provision experiment with a general population sample to see whether consumers punish companies who advertise on misinformation pages. Third, they conduct another information provision experiment with a premium sample of firm decision-makers to see how firms respond to information about misinformation advertising and their demand for information relevant to advertising and misinformation pages. Their main results show 1) that advertising on misinformation pages is surprisingly common, partly driven by digital ad platforms, 2) that consumers punish firms for advertising on misinformation pages, irrespective of whether or not firms use digital ad platforms, and 3) that firm decision-makers are often unaware of their own advertising on misinformation pages; underestimate the role played by digital ad platforms; and have a high demand for information relevant for reducing their ad exposure to misinformation sites.

Thank you for providing this great summary of our paper. We also greatly appreciate the time you spent with the manuscript.

In light of your comments, we have conducted three primary areas of revision. First, we conduct additional analyses to shed light on the consequential nature of our survey experiment with consumers. In doing so, we follow your suggestions to demonstrate the economic significance of the choices made by participants in the experiment. Second, we conducted additional analyses on the behavioral data we collected to show that using clicks on petition links in the decision-maker experiment is the outcome that is most aligned with actual signatures. We also demonstrate that different potential dependent variables give qualitatively similar results. Finally, we have provided detailed replies to each of your more specific suggestions and incorporated them into our paper. We hope that you find that this revision effectively and thoroughly addresses all suggestions.

Evaluation of the main contribution

I think this is a great paper on an important topic. I was enthusiastically reading the paper and felt I learned a lot from reading it. I also want to applaud the authors for very carefully collecting rich descriptive data on the extent of firm advertising on misinformation pages as well as conducting two information provision experiments that cover both the consumer and firm sides. The paper is also on a very important topic and, while there is not much “conceptual” novelty in terms of methods or theory development, I very much agree with the authors that the supply-side approach they push has been underexplored in the literature. While I have a lot of enthusiasm for the paper, I also have a few comments and potential criticism that I outline below.

We sincerely appreciate these encouraging words about the paper. Your comments and suggestions have been very helpful to us in revising the manuscript.

Main comments

#1. An important question is to what extent the gift card experiment is an externally valid measure of consumer responses to misinformation ads. The respondents can initially indicate their preference between six gift cards from most preferred to least preferred. Post-treatment, they can then select which gift card they actually want to receive. The main outcome is whether people choose a different gift card than their top choice (i.e., “switch”, as reported in columns 1 and 2 of Table 1). Depending on how much they value their first choice relative to the other choices, the decision to “switch” can be quite costly or not costly at all. The authors measure the weights of each gift card (which must sum to 100), so they have data on the intensity of the preference, and hence the respondents’ relative valuation of each gift card. I think it’s important to investigate how costly it was for people to switch from their top choice. For instance, they could calculate how many percent of the gift card value they lose by switching from their most preferred option to a different option (using the relative weights). E.g. if Uber gets a weight of 90 and Lyft gets a weight of 10, switching from Uber to Lyft reduces the implied gift card value much more than if Uber gets 55 and Lyft gets 45.

Thank you for this comment. Based on this, we have dug deeper to understand the cost of switching to a less preferred option. We take several steps to quantify this cost.

First, in line with your suggestion, we descriptively explore the weights assigned to the initial top choice gift card and the eventual final choice after the information treatment. Focusing on treated individuals who switched, the mean weight assigned to the initial top choice was 39.1, while the median was 35. The final gift card choice had a mean weight of 23.7 and a median of 20. This means that as a result of switching their gift card choice, the treated participants lost 39.4% to 42.8% of their initial top choice value. We believe that this magnitude is economically meaningful. Given that the value of the gift card is \$25, a 39.4% average decline means losing an equivalent of \$9.85. This decline is represented in Figure B1, which shows a significant leftward shift in the distribution of weights assigned when individuals switch away from their initial top choice gift card. Overall, this analysis suggests that our participants do lose significant value due to switching.

Figure B1: Distribution of Weights Assigned by Treated Participants to their Initial and Final Gift Card Choices

Next, we quantify this switching cost more formally by using an instrumental variable strategy to account for changes across the treatment and control groups. We run a two-stage least squares regression to analyze the impact of the information interventions on the amount of loss in terms of the weights assigned to the initial top choice and the final gift card chosen. The idea is that gift card choice is the endogenous variable and the information treatment is the instrumental variable. The treatment status does not directly affect the utility loss from switching, but it can only shift the intensity through its effect on people’s choice of the gift card option. Focusing only on our first treatment (T1) relative to control, Column 1 of Table B1 shows the first stage, where we instrument the probability of switching with the information treatment, replicating our baseline treatment effect of a significant increase in switching. In Column 2, we see that an increase in switching due to our information interventions leads to a reduction of 17 points in the weight assigned to the final choice relative to the initial top choice. In Columns 3 and 4, we run similar regressions for a composite treatment variable relative to the control group to find comparable results to Columns 1 and 2. Moreover, the estimates in Columns 2 and 4 are in line with the descriptive statistic above.

Table B1: Difference in Weight Between Final Choice and Initial Top Choice Gift Card

	First stage Switch in preference (1)	2SLS Weight difference (2)	First stage Switch in preference (3)	2SLS Weight difference (4)
(Intercept)	0.04*** (0.01)	0.37 (0.38)	0.05*** (0.01)	0.68** (0.34)
Treatment	0.13*** (0.01)		0.09*** (0.01)	
Switch in preference		-17.00*** (4.21)		-20.21*** (3.38)
Observations	1614	1614	4039	4039

*** $p < 0.01$, ** $p < 0.05$, * $p < 0.1$

Notes: This table shows IV regression results, where *switch in preference*, i.e. whether a participant changes their gift card choice post-treatment, acts as an instrument for our treatment. In Columns (1)-(2), *Treatment* is a binary variable, which takes value 1 when a respondent is randomly assigned to receive our first information treatments (i.e., T1) that mention their top choice gift card company advertising on misinformation websites. In Columns (3)-(4), *Treatment* is a pooled binary variable, which takes value 1 when a respondent is randomly assigned to receive any of our information treatments (i.e., T1, T3 and T4) that mention their top choice gift card company advertising on misinformation websites. Columns (1) and (3) show the first-stage regression result. Columns (2) and (4) show the 2SLS regression result, where the dependent variable, *Weight difference*, is the difference in the weight assigned by the participant to their final choice and initial top choice gift card. Robust standard errors in parentheses.

To illustrate the validity of the weights assigned by participants to the gift card options presented to them, we measure heterogeneous treatment effects for participants based on the difference between their highest weighted and second highest weighted gift card choice. Supplementary Table A9 shows that as this difference increases, switching decreases. This shows that the weights assigned by participants to their gift card options are capturing meaningful and costly differences in value, highing the external validity of our findings.

In addition to the above analysis, we also show in the paper that treated individuals switched to lower preference products (Columns 3-4, Table 1) after our information interventions by 8 percentage points. More generally, our pre-registered heterogeneity analysis lends credence to the study’s external validity. In line with expectations, we find that less frequent users and more politically liberal individuals are likelier to switch (see Extended Table E3 for the full set of pre-registered heterogeneity results). Moreover, we find that the cost of switching gift cards varies based on participants’ observable characteristics. For example, treated par-

participants who reported not using any of the misinformation news outlets in our survey lost 50% of the median value (\$12.50) of their initial top choice gift card whereas treated participants who reported reading such outlets lost 33.3% of the median value (\$8.33) of their initial top choice gift card.

Prior studies of customer willingness to pay for ethical products, social labels, or the impact of negative feedback provide another way to benchmark the size of our effects. For instance, previous research finds that consumers are willing to pay 10 percent less for coffee brands that do not pay farmers fair prices and include a fair trade certification label.¹⁰⁷ A meta-analysis of studies on consumers and genetically modified (GM) food products shows an average 23 to 28 percent price premium for non-GM food.¹⁰⁸ Our descriptive statistic of a 39-43% loss in value appears in line with the upper range relative to these studies. Additionally, as mentioned in the paper, the decline in demand (13 pp) is comparable to demand reduction from receiving negative product feedback.⁵⁰ It also exceeds the magnitude of demand changes associated with companies taking a political stance^{41,42} or adopting race or gender-related ownership labels.⁶¹

Overall, we hope these analyses provide a menu of evidence that illustrates the external validity of our study. We have incorporated parts of these discussions and analyses in the Average Treatment Effects Section of the paper. We hope this addresses your concerns.

Furthermore, the authors could acknowledge somewhere that they measure short-term effects and it is not clear how large the effect would be e.g. one month later when the issue was less salient. That is, it could be the case that consumers immediately punish a company for placing ads on a misinformation page, but after some time, they forget about it and it is “business as usual”. A follow-up study to test whether the effect persists would have been very interesting (in which I think Likert scale outcomes would have been totally fine).

We agree with your point that our measures are short-term in nature. Our proposed interventions mirror contexts where consumers receive information about the advertising companies they interact with. For example, information-based disclosures, as in our first treatment (T1), could be incorporated as labels similar in spirit to the “Sponsored by” and “Paid for by X” labels that are presently common on various digital media platforms.³¹ Similarly, rank-based information provided in the final information treatment (T4) could be provided as a ranking of companies where customers are selecting products from a menu of choices. Platforms do provide such contextual information about companies in other settings, e.g., Google Flights displays carbon emissions data next to each flight when people select a flight to purchase among several options. Therefore, our interventions might capture natural changes in choices at the point of purchase in response to (on-going) information provision. We now include this clarification in the Discussion Section of the paper.

That noted, in line with your point, it would have been interesting to capture longer-term effects. Unfortunately, carrying out a follow-up study was infeasible due to our budget constraints. We wanted to make the main gift card outcome as realistic as possible. Hence, we had a large value gift card (\$25) and a high probability of being drawn in the lottery (20%) relative to prior literature. The cost of implementing this

³¹<https://www.facebook.com/business/help/198009284345835?id=288762101909005>.

for a large sample and expenditure on the other studies in the paper meant that we had to forgo measuring longer-term effects since it would become cost-prohibitive. We now explicitly mention this constraint in the Methods section of the revised manuscript.

Additionally, we discuss our findings in light of the current literature examining the dynamic effects of consumer reactions to company or product information over time. Prior work shows that the impact of negative information persists for three to six months and persists longer than positive information,^{61,62} which suggests that the effects of the information interventions we propose could linger for a few months. Previous research also shows that consumer boycotts tend to reduce in intensity after eight months on average,^{109,110} but effects can persist,¹¹¹ especially if news and social media coverage increases. Given that prior literature suggests that it is the total amount of negative information that citizens accumulate that creates lasting associations,⁷⁶ providing information disclosures at scale about advertisers placing ads on misinformation websites could have a substantial effect on consumer demand over time. However, it is unclear to what degree the effects of information interventions would persist or diminish in our context. Hence, in accordance with your suggestion, in the Discussion Section of our paper, we now also highlight that studying the long-term effects of such information interventions, once provided at scale by platforms, would be an important avenue for future research.

#2. I think the authors are a bit too dismissive of the value of self-reported measures in section 3.2.4, mostly because I think the Likert scale outcome measures different things. A more direct comparison between the behavioral outcomes and self-reported/Likert scale outcomes would be along the lines of “How likely are you to use Uber in the future?”.

Thank you for this point. We agree that we should not be dismissive about the value of self-reported measures and that our Likert scale measures might be capturing something slightly different. In line with this, we have reduced the amount of space given to this point by putting it in Section Average Treatment Effects and toned down the language used in the discussion of this analysis. In particular, we write: “We also find suggestive evidence for vast differences between consumers’ stated and revealed preferences, as shown in Supplementary Figure A1. These results come with the caveat that the stated preference questions don’t map exactly into gift card or petition choices and, hence, should be viewed as suggestive.” We can remove this brief discussion and analysis from the paper entirely if you think it appropriate.

#3. The authors write that the “platform only” treatment group (T2) “practically serves as a second control group for our company-level outcomes”, yet, relative to the control group, they see a 3pp increase in their main outcome (columns 1 and 2 of Table 1) for T2 relative to the control group. I think it’s worth commenting/discussing this result in the main text.

Thank you for pointing this out. We agree that the nuance behind this is worth discussing further in the paper. The idea behind thinking that T2 could serve as an alternative control group is that in contrast to T1, T3 and T4, it does not mention that the company chosen by the respondents appeared on misinformation websites. The magnitude of switching behavior in T2 relative to our other treatments is much smaller, which bears out

this difference in treatment intensity (see Table 1 in the paper). Our control group was provided with generic information about news consumption, including misinformation, in America whereas the T2 group treatment mentions generic information about platforms, i.e. how companies that use digital ad platforms are much more likely to appear on misinformation websites. While T2 does not explicitly mention the respondents' top choice gift company (as in T1, T3 and T4) or its specific use of digital ad platforms (as in T3), T2 could still lead to some switching because survey takers might assign some responsibility to their top choice gift card company for using digital ad platforms or for not doing their due diligence after learning about how platforms place companies' ads on misinformation websites. Since their first choice gift card company might be top of mind,⁴⁸ they might blame them partially or assume that the information provided in T2 alluded to the company they had just chosen.⁴⁹ This could be the reason for the slight increase in the primary outcome variable in T2 relative to the control group. It is important to note that the other outcomes reported in Table 1 in the paper, i.e., switching to lower preference gift cards and switching across categories are not statistically significant for T2, which suggests that T2 does not result in treatment effects similar to our other treatments. We now include this discussion in the Section Average Treatment Effects.

#4. The authors use petition links as their primary "voice" outcome, which I think is a nice and relevant outcome measure. However, I think it's somewhat problematic that they focus on link clicks - which, I believe, could be quite noisy. I think the intention to sign, reported signatures and actual signatures are all preferable to link clicks (or the link clicks should be validated somehow). The authors also write that their results are robust to using petition intentions or actual signatures, but they don't have the petition signatures for the T4 treatment (the table footnote for the appendix table reveals that these petitions were accidentally deleted). I think this is worth highlighting in the main text.

We appreciate you pushing us to be more clear and to conduct additional analysis to highlight the robustness of the primary "voice" outcome. To understand which variable might best capture our voice outcome, we conduct additional analyses.

First, it is pertinent to note that we have information on link clicks, intention to sign and reported signatures available at the individual level as measured during our survey whereas actual signatures are measured at the level of the treatment group since participants in each treatment group were provided with a unique petition link. We estimate the treatment effects using each variable as a different dependent variable. For ease of exposition, we only report the results for the platform petitions in Table B2 below (with the full table included in the paper as Extended Table E2). Our results shown in Table B2 below demonstrate that using any of these three measures (i.e. reporting an intention to sign, clicking on the petition links, or self-reporting signatures) leads to qualitatively similar results. Moreover, quantitatively, the estimates across these three outcome variables are also statistically similar. Hence, the overall message of the analysis does not hinge on the particular use of any of these individual-level outcomes as the primary outcome variable.

Second, it is important to highlight the survey flow which suggests that link clicks, a binary variable that takes the value 1 when an individual clicks on the petition link provided and zero otherwise, would be the most

Table B2: Comparison of responses across platform petition outcomes.

	Intention (1)	Clicks (2)	Reported (3)	Signed (4)
Company (T1)	-0.01 (0.02)	-0.02 (0.02)	-0.02 (0.02)	-0.02 (0.01)
Platform (T2)	0.05** (0.02)	0.05*** (0.02)	0.04** (0.02)	0.04** (0.02)
Company and Platform (T3)	0.03 (0.02)	-0.01 (0.02)	0.03 (0.02)	0.00 (0.02)
Company Ranking (T4)	-0.03 (0.02)	-0.03 (0.02)	-0.03 (0.02)	
Controls	Yes	Yes	Yes	No
Control mean	0.21	0.14	0.20	0.11
Observations	4039	4039	4039	3225

*** $p < 0.01$, ** $p < 0.05$, * $p < 0.1$

Notes: This table shows OLS regression results for each of the four treatment groups (T1, T2, T3 and T4) across all of our petition outcomes. In columns (1), the dependent variable is the intention to sign a petition, a binary variable that takes the value 1 when a participant indicates wanting to sign a given petition and zero otherwise. In columns (2), the dependent variable is a click on the petition link that takes the user to the Change.org platform to sign a petition, a binary variable that takes the value 1 when a participant clicks on the link and zero otherwise. In columns (3), the dependent variable is the self-reported petition signature, a binary variable that takes the value 1 when a participant reports having signed a given petition and zero otherwise. We record actual petition signatures in columns (4). We omit signatures for the T4 group since these petitions were accidentally deleted by Change.org.

relevant outcome. After receiving an information treatment, the first voice-related outcome we measure is the intention to sign, where participants are provided with various petition options and asked if they intend to sign any of the available options. Those who indicate intending to sign a particular petition are then provided with a link to that specific petition on the next page and asked to self-report whether they signed the petition. In order to sign the petition, an individual must click on the petition link we provided in the survey. Only after clicking on the link can they access the actual petition. On the other hand, individuals can report their intent to sign or say that they signed the petition without clicking the petition link. To demonstrate this empirically, we calculate differences in the proportions of group-level actual signatures and the proportions of our three other voice-related outcomes for each treatment group. As can be seen in Figure B2 below, across different groups, the difference between link clicks and the total actual signatures is the smallest out of the three outcome variables considered, which makes our click outcome the best approximation for the actual signatures relative to the other two voice outcomes. We now include this in the paper as Extended Figure E3.

More generally, clicks have been used as a proxy outcome in studies that analyze online behavior, especially when downstream outcomes are not observable.¹¹² In online settings, clicks have been validated as a credible intermediate measure.^{113,114} In our setting, clicks on links occur right before the actual signatures, making them even more credible than standard online settings where the actual outcome of interest might be a few steps downstream, which is also borne out by Figure B2 .

Finally, in line with your suggestion, we have now highlighted in the main text that Change.org accidentally deleted the total number of actual petition signatures for T4. Hence, we do not have those numbers and cannot use them for analysis, including the results discussed above. Specifically, we write in the Section Average Treatment Effects: “It is important to note that we do not analyze signatures for the T4 group since Change.org accidentally deleted these petitions and related data.”

Figure B2: Delta Values by Treatment Group and Type of Voice Outcome.

Notes: The delta value reported here represents the difference between the proportion of a particular outcome variable and the proportion of actual recorded signatures for each treatment group.

Overall, we hope this clarifies and justifies the use of link clicks as the primary outcome variable of interest. If we have missed anything or if you believe that we should be using the other variables as the primary outcome measure, we would be happy to look into it further.

Another thing that is interesting about Table A6 is that self-reported signatures and actual signatures are almost identical (control group mean of 15% and 14%, respectively) - this actually speaks to the value of self-reported data.

We are very grateful to you for giving the paper such a careful read. We wanted to double-check these numbers based on your comment and our analysis of the different petition outcome variables above. We found a slight discrepancy in the reported control group means of the variables in the former Table A6 (now Extended Table E2), which we have now corrected. In particular, the control mean for clicks is 15% while for self-reported signatures, it is 21% and not vice versa. That is, the control means in columns (2) and (3) were switched. This was likely a manual error moving between R and Overleaf. We sincerely apologize for this and have double-checked all the numbers and results for the revised version. These numbers align with the analysis above, including Figure B2. In general, your point about self-reported measures is well taken. Indeed, our analysis shows that self-reported signatures, intention to sign, and link clicks as outcome variables lead to qualitatively similar results. As mentioned before, we have taken a more measured tone when discussing self-reported measures. Thank you for highlighting this point again.

In the discussion of Table 2, I also think the authors should discuss the null findings as well and not only focus on the statistically significant effects.

We now elaborate on the statistically significant results and include a discussion of the null effects when

discussing Table 2 in Section Average Treatment Effects of the paper. We provide a brief overview of the discussion here. In particular, we suggest that the statistically significant effect on T2 (platform only) is because the only way consumers can take action about platforms in our experiment after learning about their role in amplifying misinformation is by voicing their concerns through the relevant platforms petition. We see a null effect for company petitions in T1 (company only) potentially because the treatment is not strong enough to condemn companies in general for the misinformation problem. This logic would also potentially hold for T3 (company and platform). In T4 (company ranking), though, the treatment intensity for companies, in general, is significantly stronger since providing the ranking information highlights that all the companies we mention advertise on misinformation websites. This could lead to consumers voicing their concern about companies financing misinformation by signing Change.org petitions at significantly higher rates than the control group, as seen in Table 2. Thank you again for raising this point. We agree that discussing these null findings is important since it sheds more light on what interventions are less likely to work.

#5. Table 6 needs more documentation. The authors focus on the subsample who reported “No” to think that their companies had ads on misinformation pages and then split this sample by whether they were “certain” or “uncertain” about this. Consulting the instructions, it seems that the question of certainty was elicited on a 5-point Likert scale from “Very sure” to “Very unsure”. It is unclear whether those who were “certain” include those who said “Very sure” or also those who said “Sure”. It also leaves me a little bit uncomfortable that, splitting the sample this way, they end up with only 68 observations in columns 2/4 of Table 6 to find a 40% p.p. treatment effect. Is this effect robust to different splits of the “Sure/Unsure” variable? Was the split they decided on pre-specified? To find statistically significant effects with only 68 observations, they need really large treatment effects, which is somewhat problematic.

Thank you for pointing this out. To clarify which responses correspond to “certain” or “uncertain”, we have added the following description to Table 4: “Columns 1 and 3 show results for participants who report being certain about their response to the aforementioned question (choosing “Somewhat sure”, “Sure” or “Very sure”). Column 4 shows results for participants who report being uncertain about their response to the aforementioned question (choosing “Unsure” or “Very unsure”).” Since the mid-point of our Likert scale, “Somewhat sure”, was not neutral (e.g. Neither sure/unsure), we analyze these responses with the certain group as done in prior work.^{115,116}

More generally, we agree with your small-sample concerns. Hence, in the revised manuscript, we clearly state that these results should be considered suggestive because of the small sample and that these are exploratory because the splits were not pre-registered. We also note that the main outcomes of interest in both the decision-maker and consumer experiments along with the dimensions of heterogeneity in the consumer experiment, were pre-registered. We hope that this will give the reader proper context but still allow us to present what we think are results with meaningful insights.

To ensure that our results are robust to different sub-samples, we carry out further checks to ensure that one marginal group of individuals is not driving our results. First, we drop the most neutral (mid-point of the Lik-

ert scale) respondents who answered “somewhat sure” from the “certain group.” Table B3 demonstrates that dropping these individuals keeps the results (Column 3) qualitatively similar to the baseline results (Column 1). Similarly, we check to ensure that our results for the uncertain group are not driven by a small number of (extreme) “very unsure” individuals. Column (4) drops these individuals to find that the results are similar to the baseline in Column (2).³² Finally, as an additional robustness check, we supplement our main sample by adding the inattentive respondents to our original sample to increase the sample size. These participants were not included in our primary analysis because they failed an attention check before the information intervention. In Columns (5) and (6), we find that adding these individuals increases the sample size but has results qualitatively similar to the baseline results in Columns (1) and (2), respectively. While these robustness checks hold, this analysis comes with the same small sample caveats discussed earlier.

Table B3: Robustness Check for Treatments Effects on Platform Solution Demand Based On Prior Beliefs

	Original sample		Reduced sample		Extended sample	
	Certain (1)	Uncertain (2)	Certain (3)	Uncertain (4)	Certain (5)	Uncertain (6)
Treatment	-0.03 (0.05)	0.36*** (0.13)	-0.08 (0.08)	0.37** (0.17)	-0.05 (0.06)	0.29** (0.13)
Controls	Yes	Yes	Yes	Yes	Yes	Yes
Observations	286	68	188	56	326	75

*** $p < 0.01$, ** $p < 0.05$, * $p < 0.1$

Notes: This table shows OLS regression results for the sub-sample of participants who reported “No” to the question “Do you think your company or organization had its ads appear on misinformation websites during the past three years (2019-2021)?”. The dependent variable is the demand for platform solution (as shown in Tables 3 and 4). Columns (1) and (2) show results based on our sample split reported in the paper (Table 4) for participants who report being certain (i.e., choosing “Somewhat sure”, “Sure” or “Very sure”) and participants who report being uncertain (choosing “Unsure” or “Very unsure”) about their responses to the aforementioned question, respectively. In Column (3), we show results for the *Certain* group based on a new sample split for robustness, whereby we categorize participants as being certain if they choose “Sure” or “Very sure” in response to the aforementioned question. In Column (4), we show results for the *Uncertain* group based on a new sample split for robustness, whereby we categorize participants as being uncertain if they choose “Unsure” in response to the aforementioned question. Columns (5) and (6) replicates the OLS regressions in Columns (1) and (2), respectively, for all participants, including those whose responses to our survey questions suggested that they were inattentive during our survey. Robust standard errors in parentheses.

Minor comments

#1. On p. 13, the authors write that the treatment effect of T4 amounts to “seven percentage points”, yet the number in columns 1-2 of Table 2 shows an 8 pp change. This discrepancy should be corrected (I assume the table version is correct).

Thank you for pointing this out. We have corrected it. Indeed, the estimate reported in Table 1 was correct.

#2. In section 4.1.1 “Setting and sample size”, the sample size is actually not included.

Thank you for pointing this out. We have included the sample size of respondents in the Methods, ‘Decision-maker experiment research design’ Section of the paper.

#3. In Table 1, I would include p-values for equality of treatment effects such as T2=T3, etc.

³²Note that the treatment magnitudes are marginally different from the initial submission (40 pp vs. 36 pp) because we collected new externally verified data on job titles based on R3’s request, which we use as a control variable instead of our previous self-reported data.

We have now included these p-values in Table 1 of the paper.

#4. In section 3.2.2, the authors report percent instead of percentage points change (36% more...) I think it makes sense to also include the pp change in the main text for better context.

Thank you for this suggestion. We have included the percentage points change alongside the percent change that was reported in Section 3.2.2 of the previous manuscript in the main text of subsection Average Treatment Effects under ‘Measuring consumer responses to advertising on misinformation sites’.

#5. Footnote 23 on p. 20 says that they dropped respondents who answered greater than 100 when asked to estimate a number out of 100. I would clarify whether this was done pre-treatment or post-treatment (post-treatment is problematic because the treatment could theoretically induce people to become more/less attentive).

The questions where respondents were asked to estimate a number out of 100 correspond to the prior belief questions that preceded our information treatment. We have added this clarification to the aforementioned footnote in Section Methods, ‘Decision-maker experiment research design’, which now states: “We dropped responses where participants provided an answer greater than 100 when asked to estimate a number out of 100 in the two questions eliciting their prior beliefs about companies and platforms before the information treatment was provided.”

Thank you again for your careful read and helpful suggestions! We greatly appreciate your detailed comments, which aided us in strengthening the paper. We hope that our revisions have been a valuable addition to the paper.

Reviewer 2

This paper evaluates the possibility of a supply-side intervention to suppress the availability of misinformation (really: low quality websites) on the internet. It evaluates this with a 3 prong study. The first study evaluates the prevalence of advertising (and evaluation of which advertisers) on misinformation websites. The second evaluates whether consumers have a material preference for companies that do not advertise on misinformation websites. And the third evaluates whether decision makers at companies were aware of and would have a preference for avoiding advertising on such websites.

Overall, I quite like this paper. The issue of the financial undergirdings of misinformation sites is critical, but understudied. There is far more research on how to reduce demand for misinformation; yet it's not clear to me that many of those approaches scale. The potential for eliminating the business model for the production of misinformation is thus quite attractive as an approach to the broader issue.

Thank you so much for this positive assessment. We are glad to hear that you agreed that the issues we are exploring are critical and understudied, as well as the assessment that this is an attractive approach to the broader issue. We truly appreciate your input.

Based on your comments, the primary areas of revision we have included are threefold. First, we have revised the manuscript to incorporate nuances of experimenter demand effects and details about our pre-registration. Second, we conduct additional robustness checks to explain our results regarding the use of digital ad platforms. Finally, we include our detailed replies to each of your suggestions in the letter below, and provide the data and code as part of the revised submission. We greatly appreciate the time you spent with the manuscript and hope that you find that this revision effectively and thoroughly addresses all of your suggestions.

The three independent analyses provide a compelling package. The first study uses a data set of ads scraped from a large set of misinformation sites, finding a wide array of advertisers present, and apparently largely driven by a small number of advertising platforms. These platforms separate buyers of ads from their placement on particular websites (though there are underutilized tools to exclude websites from particular campaigns). The second finds that for a representative sample, there are material preferences for avoiding products from companies that advertise on these websites. This is done through an information intervention, where people can choose among a set of gift cards to receive, and are then informed about whether the company associated with the gift card had advertised on misinfo sites (all had, but in varying amounts). The bottom line is that the intervention drove people away from companies that advertised on the misinfo sites, as well as to sign an online petition (there are a number of interesting nuances). The third takes a sample of business leaders (alums of Stanford and Carnegie Mellon exec ed), surveys them (and includes an information intervention there as well), finding strong preferences for ads not appearing on misinfo website (again, interesting nuances).

Thanks for this great summary of the paper. We appreciate the positive words about the studies we carried out and are especially encouraged to hear that you felt our analyses provide a compelling package.

There are limitations, of course. The second and third studies have potential generalizability issues, in terms of whether this would truly hold in the field. But, I think they've taken steps to maximize the potential external validity, like make sure there were real (if not large) stakes for the subjects.

Thank you for this point. We tried to follow best practices and use appropriate incentives given our budget constraints to make our experiments as real as possible for the survey participants.

I do have concerns in the third study that the alums might want to “do the right thing” for a study from the institutions that they had attended (such as attending is for exec ed, but still).

This is a pertinent point. While the experimental literature suggests that experimenter demand effects are limited in online survey settings like ours,⁹¹ we take several steps to ensure truthful reporting in this study. We followed best practices¹⁰⁶ by keeping the treatment language neutral and ensuring the anonymity of the participants wherever possible. We used behavioral measures for our baseline and experimental outcomes, which were incentive-compatible in that participants' choices represented real trade-offs in the information they wanted to receive or in their time spent. That noted, we cannot completely rule out the role of participants wanting to “do the right thing” as at least a partial motivation for the participants' responses. We do not want to overclaim the validity of our results and seek to provide proper caveats. Hence, in the Methods Section for this study, where we provide more details about our approach in “Dealing with Experimenter Demand Effects and External Validity Challenges”, we write the following: “These design choices are made to minimize demand effects on our main outcomes of interest. However, it is possible that these effects are still relevant, partially because participants may have an interest in ‘doing the right thing’ on a survey administered by an institution they have a connection with.” Additionally, in the same Section, we measure the relevance of potential demand effects using an open-ended survey question asking participants about the purpose of the study. We find that participants' treatment status does not predict their responses to this open-ended question which suggests that, although experimenter demand effects may still be present, our overall experimental findings are not driven by these effects.

For study 1, I'd like more explanation of the 10:1 figure re use of advertising platforms (a figure that is then used in the experimental studies, so it's especially important). For example, couldn't the 10:1 figure simply reflect that the companies that used the platforms just advertised more that week? I'm assuming not, but a little more explanation in interpreting these numbers would be helpful.

The 10:1 number is based on the advertising activity by the top 100 advertisers in our sample. As described in the paper, this is a descriptive statistic computed based on the likelihood of appearing on misinformation websites when advertisers use digital ad platforms relative to when they do not. As you mention, it could be that there are unobserved time-specific factors that might drive advertising behavior. To dig deeper and rule out such alternative explanations, we build a panel dataset at the advertiser-week level from our dataset of the top 100 most active advertisers' behavior between 2019 and 2021.

In Table B4, we report the probability of a company’s ads appearing on misinformation sites based on logistic regressions in columns (1)-(3). The primary variable of interest is digital ad platform usage, which equals one when an advertiser uses a digital ad platform in a given week and zero otherwise. These regressions give us an odds ratio that has an interpretation in line with the 10:1 summary statistic we reported. When we account for time-specific factors, such as seasonality in ad spend, as captured by month fixed effects, we find that the probability of appearing on misinformation websites is approximately 29 times higher when an advertiser uses a digital ad platform (Column 1). Next, we additionally account for factors that might affect advertising companies within an industry using industry fixed effects to find that the probability is still approximately 22 times higher (Column 2). We further also interact the two sets of fixed effects to find that the probability remains high at approximately 30 times and is statistically significant (Column 3). Moreover, in Columns (4)-(6), we use a Poisson regression with the dependent variable as the number of misinformation websites a company appears on in a week. We find that the incidence ratios, with a similar interpretation as odds ratios in the binary case, are high, ranging from 23.2 in Column 4 to 16.8 in Column 5 and 17.0 in Column 6.

Table B4: Top 100 advertisers’ activity on misinformation websites

	Appears on misinformation websites			Number of misinformation websites		
	(1)	(2)	(3)	(4)	(5)	(6)
Digital ad platform usage	3.36*** (0.06)	3.07*** (0.06)	3.41*** (0.08)	3.14*** (0.07)	2.82*** (0.07)	2.83*** (0.08)
Odds Ratio / Incidence Rate Ratio	28.71	21.63	30.35	23.19	16.76	17.03
Month FE	Yes	Yes	Yes	Yes	Yes	Yes
Industry FE	No	Yes	Yes	No	Yes	Yes
Month FE × Industry FE	No	No	Yes	No	No	Yes
Number of Observations	11, 969	11, 969	11, 969	11, 969	11, 969	11, 969

*** $p < 0.001$, ** $p < 0.01$, * $p < 0.05$, $^{\circ}p < 0.1$

This table shows regression results for the top 100 most active advertisers in a given year based on weekly data collected for the years 2019 to 2021. In Columns (1)-(3), we show results from logit regressions where the dependent variable is *appears on misinformation websites*, a binary variable that takes value 1 when an advertiser appears on a misinformation website in a given week and zero otherwise. In Columns (4)-(6), we show results from poisson regressions where the dependent variable is *number of misinformation websites*, a continuous variable for the number of misinformation websites an advertiser appears on in a given week. *Digital platform usage* is a binary variable that takes value 1 when a company is using digital ad platforms in a given week and zero otherwise. Robust standard errors in parentheses.

Overall, these robustness checks suggests that time-varying factors such as advertising activity in a given month or an advertiser’s industry do not drive our descriptive statistic. The descriptive statistic has the benefit of being more easily interpretable and understandable by the general, representative consumer in our sample. Moreover, our descriptive statistic falls on the lower end of the estimates reported in Table B4 that capture the propensity to appear on misinformation websites when companies use digital ad platforms. We include this table as Extended Table E1 in the paper.

Other bits of housekeeping: it would be useful to know if there were any deviations from the PAPs (usually there would be; and this is especially important context for interpreting some of these heterogeneous treatment effects—were all of them preregistered?).

The primary outcome variables across the experiments were pre-registered. Moreover, the dimensions of

heterogeneity analyzed in the consumer experiment were also pre-registered. The heterogeneity analysis in the decision-maker experiment was more exploratory. To ensure readers have proper context for this analysis, we explicitly note this in the relevant sections of the paper.

Also, it is not clear why the analysis code would not be made available along with the data (as compared to upon request).

We apologize for this oversight. The analysis code for the paper has been made available along with the data from both experiments. We cannot publicly post the data used in the descriptive analysis because it is proprietary.

We are very thankful for each of the recommendations you provided, and we have incorporated them in revising our paper.

Reviewer 3

This manuscript provides several pieces of novel, rigorous and useful evidence on the understudied "supply side" of online misinformation. Each one of them would be interesting on its own, and in combination they present a fairly coherent picture of platforms' role in serving ads on unreliable websites, (potential) consumer backlash, and low awareness on behalf of corporate decision-makers. The descriptive evidence is comprehensive and clear, and the experiments are designed with a great deal of care to guard against cheap talk, demand effects, etc.

Thank you so much for this positive assessment and the time you spent with the manuscript. We truly appreciate it. We are particularly encouraged to hear that you felt that the combination of evidence presented a coherent and comprehensive picture.

Based on your suggestions, we include the following primary areas of revision. First, we collect new data to verify the job titles of the respondents in our decision-maker sample and examine the validity and generality of our results from this sample. Second, we have revised the manuscript to engage more directly with the current policy discussions, especially highlighting ways our recommendations could be potentially enforced. Finally, we provide our detailed replies to each of your comments in the point-by-point response below, where we clarify the terminology used and conduct additional robustness checks based on your comments. We hope you agree that these revisions make our paper stronger.

I have three general points about the paper. First, it seems to have been written primarily with an economics audience in mind (looking at citations etc.), which would be fine for an econ journal but not Nature. (Relatedly, the main text is far too long and detailed for a Nature-style article.)

Thank you for this important comment. We have taken two steps to address this concern. First, we have revised the manuscript per Nature's style and guidelines, making it shorter and different from how articles are written for economics journals. For example, the details about the experimental implementation are now in the Methods Section and several more detailed empirical checks are in the Supplementary Information file. Second, we have also gone through our reference list and ensured more balance in citations across different fields in line with the broad readership that Nature attracts. We do this also by incorporating the references suggested to us by the reviewers during this process. We hope this revised manuscript will be to your satisfaction.

Second, the weakest link here is the "decision-maker" experiment, which – while carefully and cleverly designed – has questionable generalizability due to the peculiarities of the sample.

We understand the generalizability concern associated with our second experiment sample. To investigate this further, we took several steps to benchmark our sample against relevant data and prior literature.

First, we check how well the industries of our surveyed companies in the decision-maker sample overlap with the industries of the advertising companies in our descriptive analysis. We find that all of the industries

(23 industries) we investigate for advertising on misinformation websites in our descriptive data sample are represented in our decision-maker sample. This suggests that the companies in our experimental sample span a range of relevant industries.

Second, we investigate some observable characteristics of our decision-maker study sample. As shown in Figure B3, our sample of participants represents a broad array of company sizes and experience levels at their current roles. Additionally, about 22% of the executives in our sample (and 25% of all our participants) are women, which is aligned with the 21% to 26% industry estimates of women in senior roles globally.^{103,104}

Figure B3: Characteristics of the Decision-maker Sample

Finally, we made several design decisions to ensure our sample was as generalizable as possible. For instance, partnering with two university programs instead of a specific firm allowed us to access a more diverse sample of companies than prior work that sampled specific types of firms, e.g. innovative firms, startups or small businesses.¹⁰⁰⁻¹⁰² Surveying Executive Education alumni provided access to senior managers and leaders, enabling insights into strategic decision-making, in contrast to studies relying heavily on MBA students for understanding decision-making in various contexts such as competition, pricing, strategic alliances and marketing.⁹⁶⁻⁹⁹

We include these analyses and discussions in the Methods, 'Decision-maker experiment research design' Section of the paper. That noted, we understand your broader point entirely and note this limitation about the sample explicitly in the Discussion Section. We hope this will give readers the proper context on how we view the results. We appreciate your comment that the experiment was carefully and cleverly designed and appreciate you pushing us to be more nuanced in interpreting the generalizability of our findings.

Third, the policy recommendations that follow from this last experiment seem inadequate to address the problem. For example, the authors begin by summarizing the "significant social consequences" of misinformation, but the paper ends with a call for transparency and voluntary self-regulation, perhaps brought on by internal employee pressure. As with much of the literature in this space, there is a mismatch between the stated severity of the problem and the proposed solutions.

Thank you for encouraging us to think harder about the policy recommendations of our study and their enforcement. We appreciate your input on the severity of the problem and the need for more adequate solutions. Our reading of the policy developments for countering misinformation is that academics and policymakers alike are often hesitant to push for explicit government regulation because of free speech protections for the news media. A natural result of such hesitancy is the call for self-regulation, which, as you suggest, may not be sufficient in curbing the social consequences of misinformation. In contrast to other approaches, our approach attempts to leverage information provision to mitigate the production of misinformation. The enforcement of our proposed interventions is a crucial point and may require regulatory action or at the very least a strong threat of it, as the literature on self-regulation suggests.¹¹⁷ We sincerely appreciate your suggestion of ensuring that our proposed solutions match the severity of the problem being addressed. We delve deeper into this in the specific response below by connecting our results to the current policy paradigm. We also note in the Discussion section that we will unlikely find a single solution to such a complex problem, yet we see our proposed supply-side approach as an important step in addressing it.

On the second experiment: I am skeptical that the sample comprises actual "decision-makers" at these companies. Do we know that people honestly self-reported being a CEO? Is there a way (e.g., LinkedIn) to bolster this claim? Otherwise another term should be used to describe the sample.

Thank you for raising this very relevant point. In order to address your concerns, we took several steps to dig deeper into our sample for the second experiment.

(1) First, we collected more data on the individuals participating in the second survey experiment. For the subset of participants who opted to provide their emails and company names as part of our survey, we collected data on their job titles at the time of the survey using publicly available information. Additionally, we collaborated with our partner Executive Education programs at the Stanford Graduate School of Business and Heinz College at Carnegie Mellon University to verify the job titles of our participants in a privacy-preserving manner. Using both these methods, we were able to individually verify the job titles of 316 of our 442 survey participants ($\approx 71\%$) as reported on their online profiles (e.g., LinkedIn, Pitchbook, Crunchbase, etc.) or their company websites at the time of survey administration (i.e. December 2022). Of these 316 people, we found that less than 1.3% of the individuals reported an inflated job title in our survey. These include three people in managerial and senior managerial roles who reported being an executive in our survey and one engineer who reported being a manager in an engineering or technical field. Hence, the additional data we collected suggests that individuals largely reported their job titles honestly during our survey.

(2) Next, we investigate the externally reported job titles of these individuals based on the additional data we collected. Figure B4 below (also included as Extended Figure E4 in the paper) shows a plot of these job titles, depicting the frequency of their appearance in our verified sample. By manually grouping the job titles into different occupational categories using the U.S. Bureau of Labor Statistics (BLS) definitions (available here: https://www.bls.gov/oes/current/oes_stru.htm), we find that about 94% of the individuals whose job titles we were able to verify in Step 1 are either executives or managers (i.e. corresponding to the BLS cate-

gories of 1) Top Executives, 2) Advertising, Marketing, Promotions, Public Relations, and Sales Managers, 3) Operations Specialties Managers, or 4) Other Management Occupations.). This is a very high proportion of individuals who are in decision-making positions within their companies at the time of our survey. The remaining approximately 6% of this sample consists of individuals in other roles within companies who participated in the Executive Education program, which is aimed at teaching personal or organizational leadership skills to help participants make an impact in their organizations. Organizations typically pay for these courses on behalf of their employees to develop leadership and managerial skills. These individuals could credibly make decisions or at the very least influence decision-making within their companies. We include the discussions above in the ‘Examining decision-maker beliefs and preferences’ section of the paper.

Figure B4: Verified job titles of participants in our second survey experiment by category.

(3) To check that the small sample of individuals in non-executive and non-managerial roles does not drive our results, we re-run our main regression specifications after dropping these individuals from our full sample of participants. The remaining sample consists of respondents who we verified as having decision-making roles or those who self-reported as being decision-makers. Moreover, the verified and self-reported decision-makers are similar across observable characteristics as reported in Supplementary Table A17, suggesting limited selection in our verification process. Our regression estimates in Tables B5 and B6 show that the results remain qualitatively and quantitatively similar after the exclusion of a small sample of individuals in non-executive and non-managerial roles. This check highlights the robustness of the results that we now discuss in the Methods section (Decision-maker experiment research design, ‘Dealing with Experimenter Demand Effects and External Validity Challenges’) and include these results as Supplementary Tables A14 and A15 as well.

(4) Finally, in addition to these checks, we would be happy to use an alternative term to label the sample of participants for our second experiment based on your recommendation. For example, we could call it the “corporate” or “managerial” sample in line with prior work that has used Executive Education alumni and

students as part of their analysis sample.^{96,97,99}

Table B5: Average Treatment Effects of Information Intervention for Sub-sample.

	Posterior belief			Platform solution demand		
	All (1)	Yes (2)	No (3)	All (4)	Yes (5)	No (6)
Treatment	52.10*** (15.73)	-14.69 (54.67)	61.93*** (17.26)	-0.04 (0.05)	-0.15 (0.15)	-0.04 (0.06)
Controls	Yes	Yes	Yes	Yes	Yes	Yes
Observations	395	80	315	395	80	315

*** $p < 0.01$, ** $p < 0.05$, * $p < 0.1$

Notes: This table shows OLS regression results corresponding to Table 3 in the paper for the sub-sample of participants in managerial and executive roles. To construct this subsample, we drop non-executive and non-manager participants for the 315 participants where we have verified job titles. For the remainder participants with only self-reported titles available, we drop participants who self-reported a role other than an executive or managerial role.

Table B6: Treatments Effects Based On Prior Beliefs for Sub-sample

	Posterior belief		Platform solution demand	
	Certain (1)	Uncertain (2)	Certain (3)	Uncertain (4)
Treatment	54.53*** (19.12)	132.14** (60.29)	-0.04 (0.05)	0.35** (0.13)
Controls	Yes	Yes	Yes	Yes
Observations	252	63	252	63

*** $p < 0.01$, ** $p < 0.05$, * $p < 0.1$

Notes: This table shows OLS regression results corresponding to Table 4 in the paper for the sub-sample of participants in managerial and executive roles. Similar to Table B5 above, we construct this sub-sample by dropping non-executive and non-manager participants for the 315 participants where we have verified job titles. For the remainder participants with only self-reported titles available, we drop participants who self-reported a role other than an executive or managerial role.

I'm also concerned about possible inattentiveness driving some of the puzzling results (such as the null on donation preferences). I don't see that attention screeners were used for this experiment – is that the case?

To ensure that potential inattentiveness did not drive our results, we did use an attention check. In particular, we note in the Methods, 'Decision-maker experiment research design' section that we removed respondents from the analysis who provided an answer greater than 100 when asked to estimate a number out of 100 in two of the descriptive questions preceding our information treatment. Doing so ensures that we are capturing respondents who carefully read the survey instructions and makes it less likely that our results are driven by inattentiveness.

Our donation outcome measures the proportion of respondents who prefer that we donate to the Global Disinformation Index (GDI) instead of DataKind. There are a few possible explanations for this null result. Our first behavioral outcome, *platform solution demand*, is related to information respondents could apply directly within their organization. The donation outcome could be perceived as a matter of personal preference and less relevant to the needs of the respondents' organizations. Second, given that both GDI and DataKind have similar goals of advancing technology's ethical and responsible use, respondents may have considered their missions interchangeable, leading to the similar take-up of both options. We have included this discussion of

the null result for this outcome in the ‘Information intervention results’ section.

On the policy discussion: Are the respondents in the "decision-makers" sample actually empowered to make decisions, or are they simply another set of people (i.e., employees) who can exercise their voice within a company?

As mentioned in the response above, based on the additional data on respondents’ job titles that we collected and analyzed, we find that 94% of the individuals whose job titles we verified from externally reported sources (e.g., LinkedIn, Pitchbook, Crunchbase, company websites, etc.) are either executives or hold managerial positions within their companies. Hence, we believe that they are empowered to make decisions within their companies. The remaining individuals could also influence decision-making within the company using their voice, especially given their participation in Executive Education program(s) to learn leadership or managerial skills, which is often financially supported by their company itself. We have included this point in the ‘Examining decision-maker beliefs and preferences’ section of the paper.

That aside, I am somewhat puzzled that the paper simply stops here – in other words, let’s disseminate accurate information about the role of ad platforms to corporate executives, hope they do the right thing, and call it a day. As an example of what I mean, the Digital Services Act just came into force across the EU and could have enforcement mechanisms to make exactly these kinds of changes, so it’s surprising to see no engagement with this or other relevant policy developments around the world.

Thank you for bringing up this important point and pushing us to further address our study’s policy implications. Our paper makes two policy recommendations based on our two information-provision experiments. These include 1) improving transparency for advertising companies about which sets of websites their ads appear on so they can make decisions consistent with their preferences and 2) informing consumers about which companies advertise on misinformation websites at the point of purchase. While we observe that decision-makers have a high baseline demand for learning about avoiding misinformation websites, our consumer experiment shows that businesses can face a substantial decline in demand for advertising on misinformation once consumers are informed about this phenomenon. Therefore, our second policy recommendation involves informing consumers as a means to further incentivize advertising companies to reduce the financing of misinformation.

The enforcement of these policy recommendations is a crucial point. While our experimental findings show that providing transparency about the actors financing misinformation via information disclosures and rankings could incentivize advertisers to reduce the financing of misinformation, platforms may not have sufficient incentives to self-regulate and implement these recommendations. We now include a discussion of potential enforcement mechanisms in light of existing policy developments in the Discussion Section of our paper as follows:

Our proposed interventions serve to ensure that both consumers and advertisers are provided with information about the consequences of their respective purchasing and ad placement decisions so that they can make choices consistent with their preferences. Having access to such information

is necessary for an efficiently functioning economic system in accordance with the first fundamental theorem of welfare economics. However, while digital platforms are uniquely well-positioned in the ecosystem of consumers, advertisers, and publishers to implement our proposed information interventions in the form of disclosures and rankings,⁶³ they may not have incentives to implement these recommendations. In the backdrop of mounting pressure from advertisers^{29,30,64} and calls for transparency in the programmatic ad business,^{65,66} our proposed interventions could be incorporated into existing legislation to improve transparency. These include efforts like the EU Digital Services Act, which includes a Code of Practice on Disinformation with enforceable provisions for different players in the advertising ecosystem to collectively fight misinformation, and U.S. bills like the Honest Ads Act and the Competition and Transparency in Digital Advertising (CTDA) Act, which include provisions to improve transparency in political advertising and the digital ad ecosystem in general. Importantly, in recent years, policy proposals aiming to reduce the prevalence of misinformation such as Australia's Combating Misinformation and Disinformation bill and Germany's bill against fake news have faced backlash over posing risks to free speech.^{67,68} While such proposals all face the challenge of striking the right balance between combating misinformation and protecting freedom of expression, our proposed information interventions could help counter the financial incentive to produce misinformation in the first place by reducing the unintended ad revenue going towards misinformation outlets. There are many parallels for regulation by information provision to address externalities in other industries, including chemicals (toxic release inventory reporting requirements), automobiles (miles per gallon information), food (nutrition and content labels), and airlines (greenhouse gas emissions), of which several have been demonstrated to be effective in prior work⁶⁹⁻⁷¹.

We believe incorporating these potential enforcement mechanisms and discussing our proposed interventions in the backdrop of existing policy approaches makes the paper's implications more relevant to the policy debate on mitigating misinformation. We hope you will agree.

Finally, a note on terminology. A lot of literature in this space uses terms like "unreliable" or "untrustworthy" to describe websites like those on the NewsGuard list. I'd strongly recommend adopting one of these throughout. There are a number of reasons for this, but most importantly, a lot of stories published by these websites are not strictly speaking false. Since the designations are at the publisher level, it makes sense to use a term that more closely describes the verification processes and overall commitment to accuracy.

Thank you for raising this important point on terminology. As you point out, prior research that has used the NewsGuard dataset has often used the term "untrustworthy" to describe websites.^{14,78} Such research has used NewsGuard's aggregate classification whereby a site that scores below a certain threshold (60 points) on NewsGuard's weighted score system is labeled as untrustworthy. We note in the Methods, 'Collecting data on misinformation websites' section that NewsGuard classifies each website on nine different journalist criteria for transparency and credibility. Instead of using NewsGuard's overall score for a website, we use the first criterion classified by NewsGuard for each website, i.e., whether a website *repeatedly publishes false news* to

identify a set of 1,546 misinformation websites. While 94% of the NewsGuard misinformation websites we identify in this manner are also “untrustworthy” based on NewsGuard’s classification, only about 52% of the untrustworthy websites are misinformation websites or websites that repeatedly publish false news. Our measure of misinformation is, therefore, more conservative than prior work using NewsGuard’s “untrustworthy” label. As mentioned in Section Descriptive evidence, we complement the NewsGuard list with 1,869 additional websites identified via data from the Global Disinformation Index. This non-profit organization aims to identify deliberately misleading information.³³ The final part of our website data includes 396 additional websites used in prior work on misinformation.^{46,47}

Given how our data has been constructed and the different specific data sources we use, we decided to use the term “misinformation” to denote the websites in our sample. We have noted the aforementioned difference in sample construction relative to prior work in Section Methods, ‘Collecting data on misinformation websites’. Your point is valid, though, that our definitions are at the website level. If you suggest that “unreliable” or “untrustworthy” would better capture the types of websites, we can certainly change the terminology used throughout the paper.

Other points:

- 2.3.2: *Approximately 8% of companies that do not use digital ad platforms appear on misinformation websites. Can you say more about these 8% of companies? Who are they, what distinguishes them?*

This 8% figure corresponds to the average proportion of companies that appear on misinformation websites among companies that do not use digital ad platforms in a given week. On the other hand, when companies use digital ad platforms, they appear on misinformation websites (at least once) about 80% of the time. These numbers were calculated for the sample of the top 100 most active advertisers during 2019 to 2021 by calculating proportions of companies that appear on misinformation websites when using and not using digital ad platforms over each week in this period.

To examine the types of companies in terms of their advertising on misinformation websites, we can alternatively construct our dataset at the company level and record the number of weeks a given company appears on misinformation websites when using and not using digital ad platforms. Overall, we find that among all appearances on misinformation websites, 93% occur in weeks companies are using digital ad platforms. Next, to understand potential differences in companies that appear on misinformation when using and not using ad platforms, we break down our company-level sample of the top 100 most active advertisers during 2019-2021 by the companies’ use of digital ad platforms:

1. About 20% of companies use ad platforms every week and all of them appear on misinformation websites during this period (94% of the time).
2. 24% of the companies do not make use of digital ad platforms in any week during the sample period, and none of them, barring one (which appears on only one local misinformation website about 15% of the time it

³³<https://www.disinformationindex.org/mission>

advertises), appear on misinformation websites in this period.

3. Finally, about 56% of the companies both use and do not use digital ad platforms at least once during this period and appear on misinformation websites about 47% of the time on average.

The companies in all three groups above span a broad range of industries and are not distinguishable based on observable characteristics. To take a closer look at companies in the third group above that appear on misinformation sites without using platforms, we filter the companies in this group to those that both use and do not use digital ad platforms in at least a quarter of the three-year period to ensure a meaningful comparison. This results in a small subsample that constitutes 7% of the overall sample of the top 100 most active advertisers, of which all except one appear on misinformation websites at least once when not using platforms. All of these few companies belong to Media & Entertainment or Publishing industries (with one company belonging to the Online Services category). Importantly, even within this small group of companies, the majority of appearances on misinformation occur in weeks companies are using digital ad platforms. We now mention these statistics in the Descriptive evidence Section of the paper.

- *Possibly useful citation with related descriptive evidence: Bozarth, L., & Budak, C. (2021). An Analysis of the Partnership between Retailers and Low-credibility News Publishers. Journal of Quantitative Description: Digital Media, 1. <https://doi.org/10.51685/jqd.2021.010>*

Thank you for sharing this useful citation. We have added this citation³⁴ to our paper in the front end of Main section of the paper while citing prior work with related descriptive evidence. Our paper corroborates and builds on this paper and others from the literature. Our novel proprietary data allows us to analyze advertising on a large number of websites over a much longer and consistent time horizon using data that are the industry standard. Crucially, we can observe the use of digital ad platforms by companies apart from which websites their ads appear on. This allows us to understand the relative roles of different entities within the advertising ecosystem, i.e. advertising companies and digital ad platforms, in financing misinformation.

- *Useful citation for demand effects: Mummolo, J. and Peterson, E., 2019. Demand effects in survey experiments: An empirical assessment. American Political Science Review, 113(2), pp.517-529.*

Thank you for this reference on experimenter demand effects from the Political Science literature. We have added this in our Methods section while discussing potential experimenter demand effects in detail for both our experiments.

Thank you again for the detailed comments above. Your suggestions have been extremely helpful to us in revising the manuscript.

Reviewer 4

This paper explores the prevalence and impact of advertising on web pages that traffic in misinformation. Using observational data on website advertising patterns and two survey experiments – one on a sample of ordinary Americans and another on a convenience sample of advertising decision-makers from U.S. companies – the authors convincingly make the case that web advertising on misinformation websites is prevalent and can carry negative consequences for companies advertising on those web pages. At the same time, individuals involved in the decision-making process over advertising placement both underestimate the prevalence of placement of advertising on low-quality websites and are receptive to corrective information about the true level of advertising on low-quality websites. The authors therefore conclude that low-cost interventions that make explicit the prevalence and nature of advertising on low-quality websites placed through ad markets could increase the knowledge of such practices and encourage companies to reduce their financing of such advertising.

I think that this is a really great paper that should be published. Each of the constituent pieces (observational web data, survey experiment of ordinary Americans, survey experiment of decision makers) is well designed and well executed. The information treatments are well designed and the instrumentation used to gather belief and behavior data is well thought out. I especially like the behavioral measures employed by the authors (the gift-card preference measures in the ordinary Americans experiment and the petition signing in the decision maker survey, especially). These are clever and creative measures that allow the authors to get at the underlying mechanism of choice in a compelling way.

Thank you so much for these positive comments. We truly appreciate them. We are particularly encouraged to read your comments about the study being well designed, well executed, and including clever and creative measures in a compelling way.

We also greatly appreciate the time you spent with the manuscript and hope that you find that this revision effectively and thoroughly addresses your suggestions. In line with your comments, we have incorporated the following primary areas of revision in our paper. First, we have revised the manuscript to incorporate nuances of the terminology around the use of “representative” or “diverse” to describe the sample in our consumer experiment. Second, based on your recommendation, we conduct additional robustness checks and report them in the revised manuscript. Finally, we have included our detailed replies to each of your suggestions below. Thank you for your careful reading and for your insightful, encouraging and very helpful comments.

But even stronger than the constituent pieces is how the authors bring these pieces together to tell a compelling story about a matter of critical importance in current society – and provide actionable recommendations to address that problem. The observational web data sets the stage by measuring the amount and type of advertising on low-quality web pages (measured through domains identified by widely used and recognized independent sources, such as NewGuard, and leading academic research teams such as Guess et al and Allcott et al). The survey experiment of ordinary Americans estimates the larger impact of these advertising patterns on company reputation. Specifically, the experiment assesses the effect of the provision of information about both the advertising placement of companies’ individuals support (indexed by a desire to get a gift card from a particular company in a raffle) and the impact of

the use of digital ad platforms on overall patterns of advertising on low-quality websites. And the decision-maker survey demonstrates that companies place ads on low-quality websites, not because they are actively trying to do so, but instead because those decision-makers have an incorrect view of the actual state of affairs. By providing both consumers and advertisers with correct information about the nature of the advertising landscape – information gleaned through the observational studies – the authors convincingly demonstrate that the financing of low-quality websites is the result of unintended consequences driven by incomplete information. Once such information is provided, the supply of misinformation in the informational ecosystem could be reduced – and reduced substantially given the size of the effects estimated in the experiment.

We are extremely grateful for the kind words about the overall takeaways from the paper. We are especially glad to see your acknowledgment of the actionable recommendations that our paper presents.

I did have a couple minor critiques of the paper. First, the sample of ordinary Americans is not a “representative” sample. It is better thought of as a “diverse” sample (see, for example, the discussion of samples in this paper by Coppock and McClellan: <https://journals.sagepub.com/doi/pdf/10.1177/2053168018822174>). I would change the language in the paper accordingly.

Thank you for raising this important point. While recruiting participants for our consumer experiment, we asked CloudResearch to apply a census-matched template, whereby CloudResearch automatically set quotas for demographic categories proportional to the U.S. Census on three dimensions, i.e. age, gender and race.³⁴ While this method substantially raised the cost of recruiting participants in our study, our decision to do so was motivated by the desire to improve the external validity of our results to the broader U.S. population.⁸⁰ Our terminology is consistent with the use of the word “representative” to describe samples used in experimental social science research from multiple disciplines, including research articles from economics,¹⁰⁶ management,¹¹⁸ political science¹¹⁹ and general science journals such as Nature.¹²⁰ That noted, we agree that our sample is not representative on every dimension. Hence, in the revised version of the paper, we do not use “representative” as an adjective. In the Methods, ‘Consumer experiment research design’ section of the paper, we now discuss how the sample can be considered to be “representative” on key dimensions where it matches the specific demographics of the U.S. population while being an otherwise diverse sample. In particular, we write:

CloudResearch screened respondents for our study so that they are representative of the U.S. population in terms of age, gender, and race based on the U.S. Census (2020). It is important to note that while we recruited our sample to be representative on these dimensions in order to improve the generalizability and external validity of our results, our sample is a diverse sample of U.S. internet users, which is not necessarily representative of the U.S. population on other dimensions.⁸⁰

We hope this change addresses your concern. We would be happy to revisit this point about terminology if there is some nuance we might have missed.

³⁴See: CloudResearch Resources, ‘How to Gather Demographically-Representative Samples in Online Studies’.

Second, there were several points in the paper where the authors discussed discarding respondents who failed an attention check or gave numerically non-sensible answers. Discarding such respondents is not always the best strategy given patterns of attrition in surveys. For completeness, would like to see a set of robustness checks in the appendix.

We appreciate this suggestion to include a set of robustness checks. We have followed best practices for our experimental design by including attention checks that are based on unambiguous logical questions.^{121,122} In line with other studies,^{123,124} if an individual fails attention checks in the initial stages of the survey, they are not permitted to proceed with the rest of the survey. For our consumer experiment, we collected our gift card outcome and the first voice outcome (i.e. intention to sign a given petition) for all participants; inattentive respondents were screened out of the survey before being presented with links to our Change.org petitions. Based on your recommendation, we present results in the Supplementary Information file for the full sample of respondents, including inattentive participants, in Tables A4 and A5. For our decision-maker study, Tables A11 and A12 provide these robustness checks. All of our robustness checks reveal that results for the samples that include inattentive respondents remain qualitatively similar and all of our statistically significant results hold regardless of the sample including or excluding inattentive respondents.

Additionally, we provide balance checks in Tables A3 and A10 for our consumer and decision-maker experiments, respectively. These checks further suggest that our sample is balanced across different conditions and attrition does not appear to be an issue in our studies based on observable characteristics.

But these are very minor points. I very much enjoyed reading this paper and I learned a great deal. It is an exemplar of the kind of research in computational social science that can elucidate the problems of the day and I support publication.

Thank you so much for your positive disposition towards the paper. We are truly grateful.

Reviewer Reports on the First Revision:

Referees' comments:

Referee #1 (Remarks to the Author):

I want to thank the authors for responding thoroughly to my comments. I realize it was a lot of work, but my impression is that it improved the paper in important ways.

I think an especially nice addition is that they are now discussing in a more structured way how much respondents lost by not sticking with their top choice. I like that they benchmark the loss with other studies, finding it is in the upper range relative to existing studies.

In general, I think the authors implemented reasonable changes based on my comments, introducing some more nuance and caution when needed (e.g. in the context of discussing Likert scale vs. behavioral outcomes; when discussing null results; and when discussing results based on small sample cells).

I thus think the paper is ready to be published based on the changes they implemented.

It is a great paper!

Referee #2 (Remarks to the Author):

I'm satisfied with the responses to my comments and to those of other reviewers.

Question re replication data. Given that the descriptive data are proprietary, is there a way to share elements of the data when given permission by the third parties who own the data? E.g., the NewsGuard data are proprietary, but many academics subscribe to them. These data do change over time, and I'm not sure how well versioned they are. I am certain NewsGuard would allow you to share the version of the data that were used in this paper with third parties who subscribe. This solution may not be viable with the other data used (it's not clear whether the advertising data are available for purchase), but it would be good to explore options for third parties to replicate.

Referee #3 (Remarks to the Author):

The authors have undertaken a serious and comprehensive revision that addresses all of my major concerns.

I'm particularly pleased with the additional data collection and analyses done to verify the titles of subjects in the 2nd experiment sample, as well as the robustness checks. The new Figure B4 is quite convincing. I'm fine with keeping the "decision-maker" description. (I also appreciate the authors' explanation of the website scoring and am now OK with the term "misinformation websites" as

well.)

I thank the authors for their work to produce an even stronger manuscript.

Referee #4 (Remarks to the Author):

when reviewing manuscripts for which I have previously been a reviewer, I focus my comments on the points I raised in my initial review. I do not think it is fair to raise additional concerns, except under extraordinary circumstances. Thus, I reviewed this manuscript with my original review in mind.

I believe that the authors have been responsive to my review. I thought this was a strong paper before and I believe it is a strong paper now. I support publication.

Author Rebuttals to First Revision:

Referee #1 (Remarks to the Author)

I want to thank the authors for responding thoroughly to my comments. I realize it was a lot of work, but my impression is that it improved the paper in important ways.

I think an especially nice addition is that they are now discussing in a more structured way how much respondents lost by not sticking with their top choice. I like that they benchmark the loss with other studies, finding it is in the upper range relative to existing studies.

In general, I think the authors implemented reasonable changes based on my comments, introducing some more nuance and caution when needed (e.g. in the context of discussing Likert scale vs. behavioral outcomes; when discussing null results; and when discussing results based on small sample cells).

I thus think the paper is ready to be published based on the changes they implemented.

It is a great paper!

Thank you for providing a detailed and thorough review of our paper. Your suggestions and questions greatly improved our discussion of our findings and helped us put our results in context of other studies. We are very grateful for your constructive feedback.

Referee #2 (Remarks to the Author)

I'm satisfied with the responses to my comments and to those of other reviewers.

Question re replication data. Given that the descriptive data are proprietary, is there a way to share elements of the data when given permission by the third parties who own the data? E.g., the NewsGuard data are proprietary, but many academics subscribe to them. These data do change over time, and I'm not sure how well versioned they are. I am certain NewsGuard would allow you to share the version of the data that were used in this paper with third parties who subscribe. This

solution may not be viable with the other data used (it's not clear whether the advertising data are available for purchase), but it would be good to explore options for third parties to replicate.

Thank you again for providing very helpful suggestions to help us improve our paper. Your suggestion on sharing data to allow for reproducibility is a great one. We have provided a copy of the version of the proprietary misinformation domains data used in this paper to NewsGuard and the Global Disinformation Index, respectively, so that those who sign a data use agreement with these organizations can access it. We now mention in our data availability statement in the paper that we can make the data analyzing the descriptive analysis of advertising on misinformation websites available to other researchers after they obtain permission from the proprietary sources.

Referee #3 (Remarks to the Author)

The authors have undertaken a serious and comprehensive revision that addresses all of my major concerns.

I'm particularly pleased with the additional data collection and analyses done to verify the titles of subjects in the 2nd experiment sample, as well as the robustness checks. The new Figure B4 is quite convincing. I'm fine with keeping the "decision-maker" description. (I also appreciate the authors' explanation of the website scoring and am now OK with the term "misinformation websites" as well.)

I thank the authors for their work to produce an even stronger manuscript.

We are extremely grateful for your feedback and suggestions to improve the paper. Your suggestion to collect additional data on our decision-maker survey sample helped us in showcasing a clearer picture of our participants' roles. Thank you again for all of your thoughtful comments.

Referee #4 (Remarks to the Author)

When reviewing manuscripts for which I have previously been a reviewer, I focus my comments on the points I raised in my initial review. I do not think it is fair to raise additional concerns, except under extraordinary circumstances. Thus, I reviewed this manuscript with my original review in mind.

I believe that the authors have been responsive to my review. I thought this was a strong paper before and I believe it is a strong paper now. I support publication.

We sincerely appreciate and thank you for providing valuable suggestions. We are grateful for the opportunity to provide a more nuanced description of our consumer sample as well as additional robustness checks for our experiments that made our paper stronger. We are truly grateful for the time you spent reviewing and sharing constructive feedback.